



# RADIv2, an Adaptable and Versatile Diagenetic Model for Coastal and Open-Ocean Sediments

Hinne F. van der Zant[1], Olivier Sulpis[2], Jack J. Middelburg[3], Matthew P. Humphreys[4],
Raphaël Savelli[5,6], Dustin Carroll[5,6], Dimitris Menemenlis[6], Kay Sušelj[5], and Vincent Le Fouest[1]

[1]LIttoral, Environnement et SociétéS (LIENSs) - UMR 7266, CNRS-La Rochelle Université, 2 rue Olympe de Gouges, La Rochelle, 17000, France
[2]CEREGE, Technopole Environnement Arbois-Méditerranée BP 80 13545 Aix-en-Provence, cedex 04, France
[3]Utrecht University, Department of Earth Sciences, Princetonlaan 8A, 3584 CB Utrecht, Netherlands
[4]NIOZ Royal Netherlands Institute for Sea Research, Department of Ocean Systems (OCS), PO Box 59, 1790 AB Den Burg (Texel), Netherlands
[5]Jet Propulsion Laboratory, California Institute of Technology, 4800 Oak Grove Dr, La Cañada Flintridge, CA 91011, USA
[6]Moss Landing Marine Laboratories, San José State University, 8272 Moss Landing Road Moss Landing, CA 95039, USA

**Correspondence:** Hinne F. van der Zant (hinne.van_der_zant@univ-lr.fr)

**Abstract.** Ocean biogeochemistry is being altered by anthropogenic processes such as warming, acidification, eutrophication, and deoxygenation. Global-ocean biogeochemistry models are essential for investigating present and projecting future conditions, yet they often lack detailed representations of seafloor processes, despite the seafloor's important role in material exchange between the biosphere and geosphere. To improve the representation of exchange across the sediment-water interface, we present RADIv2, a flexible and computationally efficient diagenetic model designed to simulate benthic biogeochemical processes across a range of marine environments, from coastal zones to abyssal plains. RADIv2 incorporates key features such as benthic methane cycling, a hydrodynamically controlled diffusive boundary layer thickness and porewater dispersion to the original RADI model, which enhance its ability to capture sediment-water exchange under varied environmental conditions. Using RADIv2, we develop and validate a regression-based metamodel that predicts benthic solute fluxes (oxygen, dissolved inorganic carbon, and alkalinity). This metamodel provides a universal and computationally efficient alternative to full-complexity coupled water column-sediment biogeochemical models at the global scale. Ultimately, this approach improves the representation of global biogeochemical cycles in ocean models by improving the parameterization of sediment-water exchange.

## 1 Introduction

The cycling of key elements within marine environments plays a critical role in regulating Earth's climate and supporting marine ecosystems. However, biogeochemical cycles in the ocean are currently subject to a number of alterations driven by human activities. Ocean warming affects most biogeochemical processes both directly, by influencing factors such as reaction rates and gas solubility, and indirectly, through ecosystem and circulation changes (Cheng et al., 2022; Levitus et al., 2005; Middelburg, 2019). Ocean acidification, driven by increased atmospheric carbon dioxide uptake, lowers seawater pH and alters



carbonate chemistry, with cascading effects on marine organisms and geochemical cycles (Caldeira and Wickett, 2003; Doney et al., 2009; Feely et al., 2004). Eutrophication, largely caused by anthropogenic nutrient inputs from rivers, has intensified due to a three- to fourfold increase in dissolved nitrogen and phosphorus export between 1900 and 2010 (Mayorga et al., 2010; Lacroix et al., 2021). Excessive nutrient loading disrupts coastal ecosystems by fueling harmful algal blooms, hypoxia, and coastal acidification (Laurent et al., 2017; Fennel and Testa, 2019). Deoxygenation results from both ocean warming, which

reduces oxygen solubility and ventilation (Keeling et al., 2010), and eutrophication-driven oxygen depletion, leading to the expansion of oxygen minimum zones and coastal hypoxia (Diaz and Rosenberg, 2008).

Despite global efforts to mitigate the effects of climate change, this pressure is set to increase further in the coming decades (Core Writing Team, H. Lee and J. Romero (eds.), 2023). Additionally, the ability of the ocean to sequester anthropogenic carbon dioxide over long timescales has given it a prominent role in proposed marine carbon dioxide removal (mCDR) strate-

gies, such as enhancing organic carbon burial and ocean alkalinity enhancement (OAE) (National Academies of Sciences and Medicine, 2022). However, the biogeochemical impacts of these activities, as well as the responses of marine and benthic environments to future climate change trajectories, are not fully understood.

The seafloor, covering 70% of Earth's surface, plays a fundamental role in biogeochemical cycling, affecting oxygen consumption in the deep ocean (Jørgensen et al., 2022; Sulpis et al., 2023) and accounting for approximately half of global oceanic

denitrification (Middelburg et al., 1996) and calcium carbonate dissolution (Sulpis et al., 2021). Despite this importance, the processes impacting particles, mineralization, and the exchange of solutes and solids at the sediment-water interface (SWI) are often oversimplified or completely absent in global-ocean biogeochemistry models (GOBMs) (Soetaert et al., 2000; Carroll et al., 2020; Terhaar et al., 2024; Liu et al., 2007; Yool et al., 2013). These simplifications limit our ability to simulate global biogeochemical cycles and compute carbon budgets that align with observations over longer timescales (Terhaar et al., 2024).

The wide range of published diagenetic models illustrates the complex nature of benthic biogeochemical processes (Arndt et al., 2013). Marine sediments are highly heterogeneous, exhibiting variability in composition, rheology, hydrodynamics, and biology across spatial and temporal scales (Jørgensen et al., 2022; Zeppilli et al., 2016; Snelgrove et al., 2018). Coastal sediments, for example, experience higher biological activity and greater hydrodynamic forcing compared to deep-sea environments. Moreover, they receive high organic matter fluxes from different sources and are subject to strong temporal vari-

ability and seasonality (Arndt et al., 2013). This spatial and temporal variability in coastal diagenetic processes necessitates high model resolution, increasing computational cost. Deep-sea environments, while more stable, present different challenges. Organic matter fluxes are lower and the organic matter is less reactive, physical transport processes (bioturbation, bioirrigation, and advection) are weaker, and redox reactions occur over longer timescales. While this allows for coarser model resolution, it also requires accurate representation of slower diagenetic processes, such as carbonate dissolution and long-term

burial. Furthermore, sediment properties, including composition (organic matter content, mineralogy), permeability, rheological properties (permeability), bottom-water hydrodynamics (bottom currents, wave action), and benthic biology (bioturbation and bioirrigation) modulate fluxes at the sediment-water interface.

Given this complexity, a one-size-fits-all modeling approach remains difficult to achieve across diverse marine environments. This lack of universal constraints limits the transferability of diagenetic models across different marine environments. This is



particularly restrictive in regions with limited in situ data for validation and input, as well as in large-scale applications. Several generic diagenetic models have been developed to simulate seafloor processes at broader spatial scales. Entirely kinetic models, such as OMEXDIA (Soetaert et al., 1996b), are computationally efficient because they only solve reaction rates, and do not explicitly account for equilibrium constraints such as acid-base buffering, carbonate chemistry, and mineral dissolution/precipitation. More complex generic generic diagenetic models, such as CANDI (Boudreau, 1996), STEADYSED (Van Cappellen

and Wang, 1996), MUDS (Archer et al., 2002) and MEDIA (Meysman et al., 2003), incorporate both kinetic and equilibrium reactions, requiring additional computational resources.

In this paper, we present RADIv2, an efficient and versatile diagenetic model that builds on RADIv1 (Sulpis et al., 2022), originally developed for deep-sea sediments. RADIv2 improves computational efficiency and extends applicability to both coastal and deep-ocean environments. Like RADIv1, it integrates organic matter degradation (Archer et al., 2002) with a dif-

fusive boundary layer (Boudreau, 1996) and non-linear parameterization of $CaCO_3$ dissolution kinetics (Dong et al., 2019; Naviaux et al., 2019). Additionally, RADIv2 incorporates dispersive solute transport, a dynamically varying diffusive boundary layer (DBL) thickness driven by bottom-water hydrodynamics, and diagenetic reactions associated with methane cycling. The model is implemented in Julia, a free, open-source programming language that enables improved computational efficiency through just-in-time compilation (Bezanson et al., 2017). RADIv2 supports both steady-state and non-steady-state simulations

and includes an ensemble simulation framework for constraining variable or unconstrained model parameters. Another advantage of this framework is its ability to represent model outputs as statistical ranges rather than single values, better reflecting the natural variability observed in in situ measurements over time. These features facilitate its application across diverse marine environments, ranging from stable deep-sea sediments to dynamic coastal settings.

To further improve sediment representation in global ocean biogeochemistry models (GOBMs), we use RADIv2 to create

a novel regression-based metamodel designed to predict fluxes across the SWI for key solutes linked to the ocean carbon cycle (oxygen, dissolved inorganic carbon (DIC), total alkalinity (TA)), based on bottom-water temperature, current speed, calcite saturation state, and the deposition rates of particulate organic matter (POM) and particulate inorganic carbon (PIC). The metamodel is tuned and validated across diverse marine environments, ensuring applicability over a wide range of settings. By accounting for the heterogeneity and variability in sediment properties, environmental conditions and bottom-water chem-

istry, it provides a globally-applicable solution for improving sediment representation in ocean models. This approach avoids the need for computationally-expensive full-complexity sediment-ocean coupling, enabling the integration of benthic-pelagic interactions in a computationally-efficient manner.





## 2 Methods

### 2.1 Model framework

#### 2.1.1 Transport equations

RADIv2 builds on the transport equations and sediment processes implemented in RADIv1 (Sulpis et al., 2022), which follow the framework established by Boudreau (1996). These reactive-transport equations describe the rate of change of a solid or solute, providing a budget equation that computes the net contribution by the different transport mechanisms: biogeochemical reactions, advection, diffusion and irrigation (solutes). For each solute component, the reactive-transport partial differential equation is

$$\frac{\partial C}{\partial t} = \frac{1}{\varphi}\frac{\partial}{\partial z}\left(\varphi d\frac{\partial C}{\partial z} - \varphi u C\right) + \alpha(C_w - C) + \varphi\sum R_l \tag{1}$$

and for each solid component,

$$\frac{\partial S}{\partial t} = \frac{1}{\varphi_S}\frac{\partial}{\partial z}\left(\varphi_S b\frac{\partial S}{\partial z} - \varphi_S w S\right) + \varphi_S\sum R_s \tag{2}$$

where $C$ and $S$ are the concentrations of solutes and solids in mol per cubic meter of water and in mol per cubic meter of solid, respectively [$\mathrm{mol\,m^{-3}}$], t is time [a], $\varphi$ is the porosity of the sediment [$\mathrm{m^3/m^3}$], $\varphi_S$ is the solid-volume fraction $(1-\varphi)$ [$\mathrm{m^3/m^3}$], $d$ is the effective molecular diffusion coefficient [$\mathrm{m^2\,a^{-1}}$], $b$ is the bioturbation coefficient [$\mathrm{m^2\,a^{-1}}$], $z$ is the sediment depth [m], $u$ is the pore water burial velocity [$\mathrm{m\,a^{-1}}$], $w$ is the solid burial velocity [$\mathrm{m\,a^{-1}}$], $\alpha$ is the irrigation coefficient [$\mathrm{a^{-1}}$], $v_w$ is the bottom-water solute concentration [$\mathrm{mol\,m^{-3}}$] and $\sum R$ is the net reaction rate of all biogeochemical reactions for solid or solute [$\mathrm{mol\,m^{-3}\,a^{-1}}$]. The SWI is set at a sediment depth of $z = 0$ and the bottom layer is set at $z = Z$. Within these limits, the sediment is vertically discretized with a user-defined depth resolution, $dz$.

Table 1 summarizes the reactions, reaction rates, and stoichiometric relationships implemented in RADIv2, adapted from Sulpis et al. (2022) and Boudreau (1996). A more detailed description of the processes and parameters is available in Sulpis et al. (2022). The stoichiometric ratios $\frac{n}{c}$ and $\frac{p}{c}$ represent the nitrogen-to-carbon and phosphorus-to-carbon ratios in organic matter, respectively. By default, the model applies a Redfield ratio of 106:16:1 (Redfield, 1934), but these values can be adjusted to location-specific conditions or organic matter input sources. The degradation rate constants $k_{\mathrm{POC_{fast}}}$ and $k_{\mathrm{POC_{slow}}}$ describe the breakdown of fast- and slow-degrading organic matter. These rate constants are not fixed; they exhibit variability depending on the source and composition of organic matter and they are typically adjusted to represent the specific conditions of each modeled environment (Arndt et al., 2013; Kuderer and Middelburg, 2024). The rate constants for the redox reactions in RADIv2 are uniform globally and are described in Sulpis et al. (2022) and the references therein. The dissolution rate constants for calcite and aragonite ($k_{\mathrm{diss.\,Ca}}$ and $k_{\mathrm{diss.\,ar}}$) control the dissolution dynamics of these carbonates, while the precipitation rate constant for calcite regulates the rate of calcite ($p_{\mathrm{prec.\,Ca}}$) formation through precipitation. RADIv2 incorporates the non-linear kinetics for both calcite and aragonite dissolution described by Naviaux et al. (2019) and Dong et al. (2019). The terms



$f_{O_2}, f_{NO_3}, f_{MnO_2}, f_{Fe(OH)_3}, f_{SO_4}, f_{CH_4}$, represent the fractions of organic carbon degraded by each respective available oxidant.
Oxidant limitation and interactions between pathways are modeled using widely applied Michaelis–Menten-type functions, which account for the influence of oxidant availability and the potential for certain oxidants to inhibit alternative metabolic pathways (Froelich et al., 1979; Boudreau, 1996; Soetaert et al., 1996a). This approach ensures that the model captures the overlapping and dynamic nature of these processes across the sediment depth. $\Delta$TA quantifies the net change in total alkalinity associated with each reaction. $\Sigma CO_2$ represents the sum of dissolved inorganic carbon produced or consumed during the reactions.

It is important to note that the reactions presented in Table 1 reflect how they are handled within RADIv2, rather than their canonical representations. RADIv2 does not explicitly solve for $H^+$ concentrations; instead, the model tracks DIC and TA, indirectly approximating the proton balance. The proton balance is estimated using a single Newton-Raphson iteration (Humphreys et al., 2022; Sulpis et al., 2022). Consequently, reactions are expressed in terms of $CO_2$ and $HCO_3^-$ to align with model handling of DIC and TA, even though these species may not appear in the canonical representations of the reactions.



**Table 1.** Summary of reactions, reaction rates, and stoichiometric relationships implemented in RADIv2. The table includes degradation rates for organic matter, dissolution and precipitation rate constants for carbonates, and the fractions of organic carbon degraded by different oxidants, along with their impact on alkalinity and dissolved inorganic carbon concentrations.

| Reaction | Rate [mM a$^{-1}$] | $\Delta$TA | $\Delta$DIC |
|---|---|---|---|
| **Organic matter degradation** | | | |
| $CH_2O(NH_3)_{\frac{n}{c}}(H_3PO_4)_{\frac{p}{c}} + O_2 \rightarrow CO_2 + \frac{n}{c}NH_3 + \frac{p}{c}H_3PO_4 + H_2O$ | $(k_{POC_{fast}}[POC_{fast}] + k_{POC_{slow}}[POC_{slow}])f_{O_2}$ | $+n/c - p/c$ | $+1$ |
| $CH_2O(NH_3)_{\frac{n}{c}}(H_3PO_4)_{\frac{p}{c}} + 0.8NO_3^- \rightarrow 0.2CO_2 + 0.4N_2 + 0.8HCO_3^- + \frac{n}{c}NH_3 + \frac{p}{c}H_3PO_4 + 0.6H_2O$ | $(k_{POC_{fast}}[POC_{fast}] + k_{POC_{slow}}[POC_{slow}])f_{NO_3}$ | $+0.8 + n/c - p/c$ | $+1$ |
| $CH_2O(NH_3)_{\frac{n}{c}}(H_3PO_4)_{\frac{p}{c}} + 2MnO_2 + 3CO_2 + H_2O \rightarrow 4HCO_3^- + 2Mn^{2+} + \frac{n}{c}NH_3 + \frac{p}{c}H_3PO_4$ | $(k_{POC_{fast}}[POC_{fast}] + k_{POC_{slow}}[POC_{slow}])f_{MnO_2}$ | $+4 + n/c - p/c$ | $+1$ |
| $CH_2O(NH_3)_{\frac{n}{c}}(H_3PO_4)_{\frac{p}{c}} + 4Fe(OH)_3 + 7CO_2 \rightarrow 8HCO_3^- + 4Fe^{2+} + \frac{n}{c}NH_3 + \frac{p}{c}H_3PO_4 + 3H_2O$ | $(k_{POC_{fast}}[POC_{fast}] + k_{POC_{slow}}[POC_{slow}])f_{Fe(OH)_3}$ | $+8 + n/c - p/c$ | $+1$ |
| $CH_2O(NH_3)_{\frac{n}{c}}(H_3PO_4)_{\frac{p}{c}} + 0.5SO_4^{2-} \rightarrow HCO_3^- + 0.5H_2S + \frac{n}{c}NH_3 + \frac{p}{c}H_3PO_4$ | $(k_{POC_{fast}}[POC_{fast}] + k_{POC_{slow}}[POC_{slow}])f_{SO_4}$ | $+1 + n/c - p/c$ | $+1$ |
| $CH_2O(NH_3)_{\frac{n}{c}}(H_3PO_4)_{\frac{p}{c}} \rightarrow 0.5CO_2 + 0.5CH_4 + \frac{n}{c}NH_3 + \frac{p}{c}H_3PO_4$ | $(k_{POC_{fast}}[POC_{fast}] + k_{POC_{slow}}[POC_{slow}])f_{CH_4}$ | $+n/c - p/c$ | $+0.5$ |
| **Redox reactions** | | | |
| $Fe^{2+} + 0.25O_2 + 2HCO_3^- + 0.5H_2O \rightarrow Fe(OH)_3 + 2CO_2$ | $k_{Fe}[Fe^{2+}][O_2]$ | $-2$ | $0$ |
| $Mn^{2+} + 0.5O_2 + 2HCO_3^- \rightarrow MnO_2 + 2CO_2 + H_2O$ | $k_{Mn}[Mn^{2+}][O_2]$ | $-2$ | $0$ |
| $H_2S + 2O_2 + 2HCO_3^- \rightarrow SO_4^{2-} + 2CO_2 + 2H_2O$ | $k_S[H_2S][O_2]$ | $-2$ | $0$ |
| $NH_3 + 2O_2 + HCO_3^- \rightarrow NO_3^- + CO_2 + 2H_2O$ | $k_{NH}[\Sigma NH_4][O_2]$ | $-2$ | $0$ |
| $CH_4 + SO_4^- \rightarrow HCO_3^- + HS^- + H_2O$ | $k_{CH_4,SO_4^-}[CH_4][SO_4^-]$ | $+2$ | $+1$ |
| $CH_4 + 2O_2 \rightarrow CO_2 + 2H_2O$ | $k_{CH_4,O_2}[CH_4][O_2]$ | $0$ | $+1$ |
| **CaCO$_3$ dissolution and precipitation** | | | |
| $CaCO_3 \rightarrow Ca^{2+} + CO_3^{2-}$ | $[Calcite] \cdot k_{diss.\ Ca} \cdot (1 - \Omega_{Ca})$ | $+2$ | $+1$ |
| $CaCO_3 \rightarrow Ca^{2+} + CO_3^{2-}$ | $[Aragonite] \cdot k_{diss.\ ar} \cdot (1 - \Omega_{ar})$ | $+2$ | $+1$ |
| $Ca^{2+} + CO_3^{2-} \rightarrow CaCO_3$ | $[Ca^{2+}] \cdot (1 - p_{prec.\ Ca})$ | $-2$ | $-1$ |



### 2.1.2 Numerical methods and computational enhancements

The original RADIv1 framework used a forward Euler method to solve fundamental transport equations, which requires user-defined time and depth steps that must adhere to a specific ratio to ensure numerical stability (Sulpis et al., 2022). However, in coastal environments with steep solute gradients between porewater and bottom waters, as well as high microbial activity,
this numerical scheme can become unstable and computationally inefficient. Capturing the complexity of sediment processes requires sufficient vertical discretization, which increases computational demand. RADIv1 and RADIv2 are the first diagenetic models built using DifferentialEquations.jl, a high-performance and modular Julia package for solving differential equations (Rackauckas and Nie, 2017). Coastal sediments may require higher-order stiff solvers with fine-tuned tolerance settings to ensure numerical stability while maintaining computational efficiency. In contrast, deep-sea environments, where geochemical
processes evolve over longer timescales with more gradual changes, can be simulated using computationally-efficient solvers that prioritize stability over high temporal resolution. RADIv2 allows users to configure solver options, including time-stepping strategies, stiffness detection, and convergence criteria, based on the specific characteristics of the sediment environment being simulated. Additionally, DifferentialEquations.jl integrates with Julia's multi-threading and multiprocessing capabilities, allowing RADIv2 to handle computationally-intensive tasks, such as ensemble simulations, with reduced runtime. Users can
define the number of trajectories and specify parameters for each trajectory individually, either by explicitly setting values or randomizing them within predefined ranges. This ensemble approach enables a more-comprehensive exploration of model behavior, particularly in cases where input parameters are uncertain or highly variable. Instead of relying on single-point estimates, users can systematically vary key input parameters such as organic matter flux, bottom-water temperature, and calcite saturation state within defined ranges. This flexibility is especially useful in coastal environments, where such parameters exhibit significant spatial and temporal variability. Additionally, this capability can be used to constrain unknown parameters by
comparing simulation results with observational data. The resulting model outputs form an envelope of potential outcomes that aligns well with in situ data, which is also typically reported as ranges due to variability within measurement periods.

Building upon an open-source programming language, RADIv2 leverages Julia's flexibility to integrate external packages, including optimization frameworks. One such package, BlackBoxOptim.jl (Feldt and Stukalov, 2018), provides an efficient,
derivative-free approach to parameter tuning. This heuristic optimization algorithm employs an adaptive gradient strategy, beginning with broader exploratory steps before refining its search space for improved convergence. This adaptability ensures optimization across a wide range of environmental conditions while minimizing the risk of becoming trapped in local minima. By incorporating BlackBoxOptim.jl, RADIv2 allows users to fine-tune model parameters to align with observational data. This feature is particularly useful for constraining the best fit to observations for highly variable or unknown model parameters.
RADIv2 includes an example script demonstrating how to optimize key parameters (https://doi.org/10.5281/zenodo.15397043; (van der Zant et al., 2025)). This script serves as a practical guide, showcasing how users can refine model outputs to match empirical data. Furthermore, this optimization framework can be extended beyond the examples provided, allowing users to fine-tune any model parameter to improve representation across different marine environments.





### 2.1.3 Method of lines and transport mechanisms

At present time, DifferentialEquations.jl only solves Ordinary Differential Equations (ODEs); however, the fundamental transport equations (Equations 1 and 2) are partial differential equations (PDEs) in both time ($\partial t$) and depth ($\partial z$). The Method of Lines (MOL) numerical technique was employed to transform the transport PDEs into a system of ODEs compatible with DifferentialEquations.jl version v7.14.0. This is achieved by discretizing the spatial domain into finite depth intervals ($dz$), while keeping time ($t$) as a continuous variable. The sediment column is divided into these discrete depth intervals and the rate of

change in solute concentrations over time is computed for each depth segment, following the approach described in Boudreau (1996).

To approximate the spatial derivatives of concentration profiles, a central difference approach is applied to the transport equations. This numerical method estimates spatial derivatives by evaluating concentration differences at two points on either side of a reference point ($z$), and assumes the concentration profile between points is approximately linear over small intervals

of $dz$. For a concentration profile ($v$, representing either a solute or solid concentration), the central difference approximation for second-order derivatives at point $z$ is:

$$\frac{\partial^2 v}{\partial z^2} \approx \frac{v_{(z+dz)} - 2v_z + v_{(z-dz)}}{dz^2} \tag{3}$$

and a backward difference scheme for first-order derivatives:

$$\frac{\partial v}{\partial z} \approx \frac{v_{(z+dz)} - v_{(z-dz)}}{2dz} \tag{4}$$

Using this central difference approach, (bio)diffusion is computed as follows:

$$D_z(v) = d_{z,T}(v)\frac{\delta^2 v}{\delta z^2} = d_{z,T}(v) \cdot (v_{(z-dz)} - 2v_z + v_{(z+dz)})/dz^2 \tag{5}$$

where $v$ represents either $C$ (solute concentration) or $S$ (solid concentration). $d_{z,T}(v)$ [m$^2$ a$^{-1}$] denotes the effective diffusion coefficient for solutes and the bioturbation mixing rate for solids ($b_z$). The bioturbation mixing rate ($b_z$) is empirically based on a 53-site data set compiled by Morford and Emerson (1999) ranging from the deep sea to the shelves [Archer et al. (2002)].

$$b_z = b_0 e^{-(z/\lambda_b)^2} \frac{[O_2]_w}{[O_2]_w + 0.02}, \tag{6}$$

where the characteristic depth $\lambda_b = 0.08$ m [Archer et al. (2002)] and $[O_2]_w$ [mol m$^{-3}$] is the oxygen concentration of bottom waters.

Advection is calculated as:

$$\text{advection} = -u\frac{\delta v}{\delta z} = \begin{cases} \text{solute} & A_z(C) = -(u_z - \frac{d_{z,T}(C)}{\varphi_z} \cdot \frac{d\varphi_z}{dz} - d_T^\circ(C) \cdot \frac{d(1/\theta_z^2)}{dz}) \cdot \frac{C_{z+dz} - C_{z-dz}}{2dz} \\ \text{solid} & A_z(S) = -(w_z - \frac{db_z}{dz} - \frac{b_z}{\varphi_{S,z}} \cdot \frac{d\varphi_{s,z}}{dz}) \cdot \frac{(1-\sigma_z)S_{z+dz} + 2\sigma_z S_z - (1+\sigma_z)S_{z-dz}}{2dz}, \end{cases} \tag{7}$$





The term $u$ in $-u\frac{\delta v}{\delta z}$ represents the effective advection term, which is modified by additional corrections to account for changes in porosity, tortuosity, and bioturbation gradients. $d_T^\circ(C)$ [$m^2\,a^{-1}$] is the "free-solution" molecular diffusion coefficient for a solute and $\theta_z$ [-] is the depth-dependent tortuosity ($\theta_z = \sqrt{1 - 2\ln(\varphi_z)}$) (Boudreau, 1996), and $\sigma_z$ [-] parameterizes influence of advection relative to bioturbation (Fiadeiro and Veronis, 1977; Boudreau, 1996). The parameter $\sigma_z$ is introduced for the advective transport of solids because a more sophisticated numerical approach is needed when biodiffusion diminishes

at depth within the sediment column. Under these conditions, the central difference scheme becomes unstable when applied to advective (burial) terms (Boudreau, 1996). A weighted difference scheme developed by Fiadeiro and Veronis (1977) is employed, that introduces the parameter $\sigma_z$ ($\sigma_z = \frac{1}{tanh(Pe_z)} - \frac{1}{Pe_z}$), which represents half of the cell Peclet number ($Pe_h = \frac{w_z \cdot dz}{2b_z}$) and measures the relative influence of advection compared to biodiffusion over a given depth interval. When $Pe_z \ll 1$, biodiffusion dominates, whereas when $Pe_z \gg 1$, advection becomes the primary transport mechanism. The scheme transitions

from a centered-difference approach under diffusion-dominant conditions to a backward-difference approach when advection dominates, and ensures numerical stability even as biodiffusion approaches zero (Fiadeiro and Veronis, 1977). This correction is not applied to solutes, where advection dominance is uncommon (Boudreau, 1996).

    The mixing of solutes caused by burrow flushing or ventilation occurs through multiple processes that are collectively termed irrigation, which is computed as

$I_{t,z}(C) = \alpha_z(C_2 - C_z),$                                                        (8)

    where $\alpha_z$ is an irrigation coefficient, approximating an ensemble of biological processes. Archer et al. (2002) derived $\alpha_z$ profile based on a 53-site dataset compiled by Morford and Emerson (1999), ranging from deep sea to shelves, expressing the irrigation coefficient as a function of the organic carbon deposition flux and the oxygen concentration of the overlaying water:

$$\alpha_0 = 11\left(\frac{tan^{-1}\left(\frac{5F_{POC}\times 10^2 - 400}{400}\right)}{\pi} + 0.5\right) - 0.9 + \frac{20[O_2]_w}{[O_2]_w + 0.01} \cdot \frac{F_{POC}\times 10^2}{F_{POC}\times 10^2 + 30} \cdot e^{\frac{-[O_2]_w}{0.01}}. \qquad (9)$$

The per-depth irrigation coefficient follows:

$$\alpha_z = \alpha_0 e^{-(z/\lambda_i)^2}, \qquad\qquad\qquad\qquad\qquad\qquad\qquad\qquad\qquad\qquad\qquad\qquad\qquad (10)$$

    where the characteristic depth $\lambda_i$ is 5 cm (Archer et al., 2002).

    The contribution of biogeochemical reactions ($\sum R$) are summarized in Table 1.


    At the sediment–water interface, RADIv2 allows for prescribed solid fluxes and regulates solute exchange through a diffusive boundary layer. Following Boudreau (1996), advection and diffusion at z=0 for both solutes and solids are calculated as follows:





$$C_{-dz} = C_{dz} + \frac{2\theta_z^2 dz}{\delta}(C_w - C_0) \tag{11}$$

and

$$S_{-dz} = S_{dz} + \frac{2dz}{b_0}\left(\frac{F_S}{\varphi_{s,0}} - w_0 S_0\right) \tag{12}$$

respectively. Here, $C_{-dz}$ and $S_{-dz}$ are imaginary concentrations used to solve transport across the SWI. For solutes, $\delta$ denotes the thickness of the DBL [m], $C_w$ is the solute concentration in the well-mixed overlaying waters [mol m$^{-3}$], and $C_0$ is the solute concentration at the SWI [mol m$^{-3}$]. For solids, $F_s$ is the particulate deposition flux at the SWI [mol m$^2$ a$^{-1}$], $w_0$ is the solid burial velocity at the SWI [a$^{-1}$], and $S_0$ is the particulate concentration at the SWI [mol m$^{-3}$].

The bottom boundary condition is characterized by a "no-gradient" constraint, meaning that the concentration remains constant at the boundary:

$$v_{Z+dz} = v_{Z-dz}, \tag{13}$$

where $v$ stands for either $C$ (solute concentration) or $S$ (solid concentration), Z is the boundary of the modeled sediment domain, $v_{Z+dz}$ is the concentration outside of the sediment domain, and $v_{Z-dz}$ is the deepest depth interval within the modeled sediment domain.

## 2.2 Additions to RADIv2

RADIv1 was designed for deep-sea sediments and does not account for methane or solute transport by bottom-water hydrodynamics (Sulpis et al., 2022). In RADIv2, we added methane to address its role in sediment biogeochemistry, particularly in coastal regions where methane formation and release can play a larger role compared to the deep sea. In coastal and shelf environments, hydrodynamic properties are highly variable and can influence key transport mechanisms, such as the thickness of the DBL (Levich and Tobias, 1963; Santschi et al., 1983; Lorke et al., 2003) and enhanced porewater diffusivity through dispersion (McGinnis et al., 2014).

### 2.2.1 Diffusive boundary layer thickness

In RADIv1, the DBL thickness was prescribed as a fixed value. However, in reality, it varies with bottom-water hydrodynamics and differs by solute due to their distinct molecular diffusion rates (Levich and Tobias, 1963; Yuan-Hui and Gregory, 1974; Santschi et al., 1983; Lorke et al., 2003). The DBL thickness is primarily controlled by bottom-water currents, ranging from tens of microns in high-energy environments (e.g., coastal areas and rivers) to several millimeters in the low-energy deep sea (Sulpis et al., 2019). The benthic O$_2$ uptake of coastal sediments can fluctuate by more than 30% within hours or days, driven





by variations in the thickness of the DBL (Glud et al., 2007). To account for this, a hydrodynamic- and solute-dependent DBL
thickness is implemented in RADIv2, following the combined approach of Sulpis et al. (2018) and Lorke and Peeters (2006).

The vertical velocity profile of bottom-water currents is approximated using the Law of the Wall (LOW) equation used in
Sulpis et al. (2018):

$$U = \frac{u^*}{\kappa} ln(\frac{\kappa u^*}{f\nu}) + Cu^*, \tag{14}$$

where $U$ is the bottom-water current velocity [m/s], $\kappa$ is the von Karman constant (0.40) [unitless], $u^*$ is the friction velocity
[m/s], $\nu$ is the kinematic viscosity of seawater $[10^{-2}\,\mathrm{cm^2\,s^{-1}}]$ and $C$ is an empirical dimensionless constant with a value of
5.1 [Monin and Yaglom (1971)]. The Coriolis parameter $f$ was approximated as $1 \times 10^{-4}$ [s$^{-1}$], a value representative of
mid-latitudes. However, $f$ is latitude-dependent, varying as $f = 2\Omega sin\phi$ where $\phi$ is the latitude and $\Omega = 7.2921\times10^{-5}$ rad/s is
the Earth's angular velocity. It is zero at the equator ($\phi = 0°$) and reaches a maximum of $1.45\times10^{-4}$ s$^{-1}$ at the poles (($\phi = 90°$)).
To assess the sensitivity of bottom-water current velocity to variations in $f$, we recalculated $U$ for latitudes ranging from $1°$
to $90°$, keeping the kinematic viscosity of seawater ($\nu$) constant at $10^{-6}$ m$^2$ s$^{-1}$ and $u^*$ at 0.1 m/s. The resulting $U$ values were
6.38 m/s ($1°$), 5.54 m/s ($30°$), 5.40 m/s ($60°$), and 5.37 m/s ($90°$). These results show that $U$ stabilizes at higher latitudes but
increases significantly near the equator due to the near-zero Coriolis parameter. Future work could improve this parameteriza-
tion by explicitly incorporating latitude dependence ($\phi$), enhancing the model's applicability across diverse geographic regions.

Figure 1b presents the empirical relationship between the kinematic viscosity of seawater ($\nu$) and bottom-water temperature
($T$, in °C), derived by fitting data from (ITTC et al., 2011). The resulting linear equation is:

$$\nu = 1.75 \times 10^{-6} - 3.24 \times 10^{-8} \times T. \tag{15}$$

By substituting this empirical relation for ($\nu$) (Equation 11) into Equation 10, we obtain an equation linking bottom current
speeds ($U$), bottom-water temperature ($T$), and friction velocity ($u^*$). This relationship, for $T = 0$ to $40°C$ and $U$ up to 1.1
m/s, is depicted in Figure 1a and is expressed in Equation 12 as

$$u^* = 0.00136 - 2.19598542 \times 10^{-5} \times T + 2.35862843 \times 10^{-2} \times U. \tag{16}$$




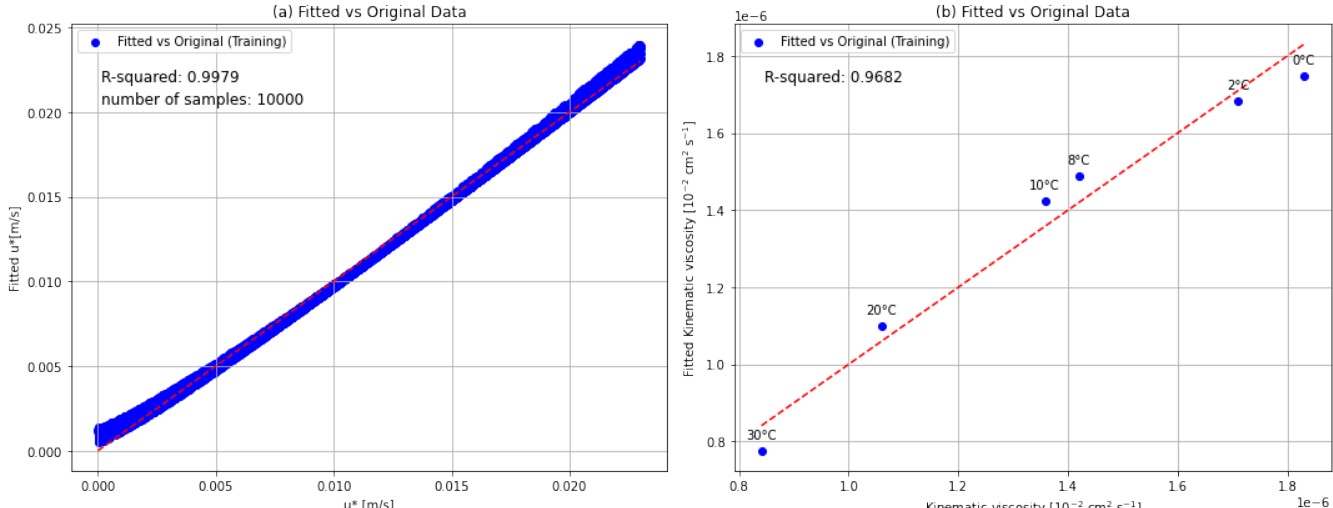

**Figure 1.** a. Fitted friction velocity ($u^*$) as a function of bottom-water temperature ($T$) and bottom-current speed ($U$) compared to the original $u^*$ values. b. Empirical relationship between kinematic viscosity ($\nu$) and bottom-water temperature ($T$), fitted to the original data (ITTC et al., 2011).

The mass transfer coefficient as a function of $u^*$ is approximated following Lorke and Peeters (2006):

$$k = \frac{1}{9} Sc^{-1/2} u^*,\tag{17}$$

where $k$ is the mass transfer coefficient [$\mathrm{m\,a^{-1}}$] and $Sc$ is the dimensionless Schmidt coefficient, defined as $Sc = \nu/d^{\circ}(v)$,

in which $d^{\circ}(v,T)$ [$\mathrm{m^2\,s^{-1}}$] is the temperature-dependent molecular diffusion coefficient of the solute.

The DBL thickness ($\delta_{DBL}$) [m] is obtained by dividing molecular diffusion coefficient ($d^{\circ}_{v,T}$) by the the mass transfer coefficient ($k$), following Sulpis et al. (2018) as

$$\delta_{DBL} = \frac{d^{\circ}_{v,T}}{k}.\tag{18}$$

**2.2.2   Pore water dispersion**

In permeable sediments (permeability $> 10^{-12}$ m$^2$ (Huettel et al., 2014)), enhanced pore water diffusivity, or dispersion, is driven by flow interaction with bottom water topography, density and pressure gradients, and migrating sediment ripples and wave motion (Boudreau, 1997; Huettel et al., 2014). The net effect on vertical transport can be represented as enhanced pore water diffusivity (Boudreau, 1997), which is estimated as a function of shear velocity ($u^*$), following the relation described in

McGinnis et al. (2014):



$$K_{v,z,T} = d_{v,z,T} e^{\alpha \kappa u^*}, \tag{19}$$

where $K_{v,z,T}$ is the apparent bulk sediment diffusivity (dispersion) [$m^2\,s^{-1}$], $d^{\circ}_{v,T}$ is the effective diffusion coefficient, corrected for sediment tortuosity [$m^2\,s^{-1}$], $\alpha$ is a sediment-permeability-dependent fit parameter [$s\,m^{-1}$] (set to 1, based on permeable sediments from the Wadden Sea and North Sea, following McGinnis et al. (2014)), $\kappa$ is the von Kármán constant [-], and $u^*$ is the friction velocity [$mm\,s^{-1}$].

For sediments with permeability below $10^{-12}\,m^2$, dispersive transport is considered negligible (Huettel et al., 2014), and $\alpha$ is set to 0, such that, in the absence of advection, only molecular diffusion governs solute transport. For permeable sediments ($> 10^{-12}\,m^2$), sediment dispersion ($K_{v,z,T}$) is incorporated as the effective diffusion coefficient ($d_{(v,z,T)}$) in solute transport equations (Equations 3 and 5), accounting for both diffusion and advection.

### 2.2.3 Methane

In RADIv1, methane, a potent greenhouse gas, was not included as a modeled solute due to its limited role in deep-sea sediments. Methane is produced in anoxic sediment layers through methanogenesis, a fermentation process that occurs once electron acceptors—such as oxygen, nitrate, and sulfate—are depleted. This pathway is primarily relevant in coastal and continental shelf sediments, where high organic matter flux supports rapid sulfate consumption and creates conditions favorable to methane production. In contrast, deep-sea sediments rarely reach the methanogenic zone due to lower organic matter fluxes and sufficient availability of favorable electron acceptors. Once formed, methane is largely consumed within the sediment by microbial oxidation before reaching the overlying water column. The primary sink for methane is anaerobic oxidation, facilitated by sulfate-reducing bacteria (Hinrichs et al., 1999; Egger et al., 2018). Aerobic oxidation can also occur near the sediment-water interface where oxygen is present, though it is typically a less significant pathway. Approximately 80% of methane oxidation occurs in continental shelf sediments (Egger et al., 2018). However, recent studies suggest that anaerobic oxidation alone may not fully counteract future methane emissions, as climate change-induced warming could outpace the capacity of benthic microbial communities to consume it (Stranne et al., 2022). The inclusion of methane in RADIv2 enables explicit simulation of methane production and oxidation pathways, providing a framework to assess how changing environmental conditions may influence benthic methane cycling and seafloor seepage.

## 3 Metamodel

Benthic-pelagic coupling plays a fundamental role in ocean biogeochemistry, influencing carbon sequestration, nutrient cycling, and climate regulation (Soetaert et al., 2000). However, many GOBMs either oversimplify or completely omit processes governing the exchange of solutes and particulates at the SWI (Carroll et al., 2020; Terhaar et al., 2024; Séférian et al., 2020). This omission can introduce biases in carbon and biogeochemical budgets, particularly in coastal and shelf regions, where sediment dynamics influence water column biogeochemistry (Middelburg and Soetaert, 2004) and thus air-sea gas exchange. The



challenge of benthic representation in GOBMs is compounded by the spatial heterogeneity of seafloor sediments (Jørgensen et al., 2022) and variability in key environmental parameters, such as organic matter fluxes and bottom-water concentrations. These complexities make it difficult to apply generic parameterizations across diverse benthic environments, while fully integrating benthic processes into GOBMs is constrained by high computational cost.

To improve the representation of benthic-pelagic coupling in GOBMs, we developed a metamodel based on RADIv2 outputs, simulating SWI exchange across six distinct marine environments: the North Sea, Monterey Bay, the shallow Iberian margin, the deep Iberian margin, and two deep-sea stations in the Arabian Sea. These sites span a range of sedimentary and hydrodynamic conditions, from productive coastal systems to deep-sea settings. The benthic flux ($J_v$) of a given solute across the SWI is driven by the concentration gradient across the DBL and follows the formulation of Sulpis et al. (2022):

$$J_v = \varphi_0 d^{\circ}_{v,T} \times \frac{v_0 - v_w}{\delta}, \qquad (20)$$

where $J_v$ [mol m$^{-2}$ a$^{-1}$] is the benthic solute flux, $\varphi_0$ [m$^3$/m$^3$] is the porosity at the SWI, $d^{\circ}_{v,T}$ [m$^2$ s$^{-1}$] is the temperature-dependent molecular diffusion coefficient, $v_0$ [mol m$^{-3}$] and $v_w$ [mol m$^{-3}$] are solute concentrations at the SWI and in the bottom waters, respectively, and $\delta$ [m] is the thickness of the DBL.

    To account for environmental variability, we performed ensemble simulations using RADIv2 that vary five key parameters:
bottom-water temperature, current velocity, calcium saturation state ($\Omega_{Ca}$), and POM and PIC fluxes. These factors were selected because they are fundamental drivers of benthic hydrodynamics, organic matter degradation, and calcite dissolution, and effectively capture the majority of the variability in the solute fluxes represented in the metamodel.

    Table 2 summarizes the key model parameters used in the ensemble runs. All run files and input files are publicly available at https://doi.org/10.5281/zenodo.15397043; (van der Zant et al., 2025).

**Table 2.** Model parameters for the six marine environments. The five varying parameters are bottom-water temperature, current speed, bottom-water calcite saturation state ($\Omega_{Ca}$), particulate organic matter flux (F$_{POM}$), and calcite flux (F$_{Ca}$). The parameters (k$_{fast}$, k$_{slow}$) represent the degradation rate constants for fast- and slow-degrading organic matter, respectively, while (p$_{fast}$, p$_{slow}$) denote the corresponding fractions of each organic matter type. $\Omega_{Ca}$ is calculated from environmental parameters using CO2SYS (Humphreys et al., 2022).

| Parameter | North Sea | Monterey Bay | Iberian margin (shallow) | Iberian margin (deep) | Arabian Sea (WAST) | Arabian Sea (SAST) |
|---|---|---|---|---|---|---|
| **Number of Runs** | 30 | 30 | 30 | 30 | 30 | 30 |
| **Run time [a]** | 1500 | 1500 | 1500 | 2000 | 2000 | 2000 |
| **Temperature [°C]** | 5.0–13.0 | 4.0–8.0 | 11.7–12.8 | 2.5–11.0 | 1.0–4.0 | 1.0–4.0 |
| **Current velocity [m/s]** | 0.02–0.10 | 0.05–0.1 | 0.02–0.08 | 0.01–0.05 | 0.02–0.06 | 0.02–0.06 |
| $\Omega_{Ca}$ **[-]** | 2.5–4.5 | 2.5–5.0 | 4.13–4.38 | 1.6–4.0 | 1.1–2.0 | 0.98–1.9 |
| **F$^{POM}$ [g m$^{-2}$ a$^{-1}$]** | 80.0–170.0 | 70–140 | 26.0–55.0 | 3.8–28.55 | 26.0–66.0 | 1.0–12.0 |
| **F$_{PIC}$ [mol m$^{-2}$ a$^{-1}$]** | 0.4–1.0 | 0.2–0.8 | 0.2–0.8 | 0.05-0.6 | 0.1–0.5 | 0.1–0.4 |
| **k$_{fast}$ [a$^{-1}$]** | 22.96 | 18.35 | 2.73 | 2.73 | 19.6 | 21.5 |
| **k$_{slow}$ [a$^{-1}$]** | 0.012 | 0.07 | 0.26 | 0.26 | 0.032 | 0.58 |
| **p$_{fast}$ [%]** | 52 | 52 | 64 | 64 | 41 | 42 |
| **p$_{slow}$ [%]** | 48 | 48 | 36 | 36 | 59 | 58 |
| **Depth [m]** | 13.5–683.313 | 100 | 104–391 | 895–4941 | 4050 | 4420 |





### 3.1 North Sea

Figure 2a shows the fluxes generated by RADIv2 alongside a comparison with values reported in the literature. RADIv2 is able to simulate steady-state benthic solute fluxes within the measurement uncertainties reported in previous studies. However, the model produces narrower flux ranges compared to in-situ measurements. This discrepancy can be attributed to the steady-state nature of the simulations, which do not account for transient variations such as temperature, tidal oscillations and internal waves. Internal waves, including their breaking counterparts, can induce sediment resuspension (Stastna and Lamb, 2008; Hosegood and van Haren, 2004), which in turn affects benthic fluxes (Niemistö and Lund-Hansen, 2019) and influences the benthic boundary layer through turbulence (Diamessis and Redekopp, 2006). Tidal cycles can also introduce substantial short-term variability in fluxes (McGinnis et al., 2014). These dynamic processes, which are not captured by steady-state simulations, can lead to significant fluctuations in solute and particulate fluxes across the sediment-water interface, particularly in coastal regions where wave action and tidal forcing are more pronounced.

The modeled remineralization rates (Figure 2b) are within the same order of magnitude than those observed in continental shelf environments, although they exceed observations from the North Sea. The relative contributions of aerobic and anaerobic remineralization in RADIv2 align with both reported values for North Sea sediments and typical continental shelf environments (Jørgensen, 1982) (Figure 2b). This agreement indicates that RADIv2 captures organic matter degradation pathways under varying redox conditions.



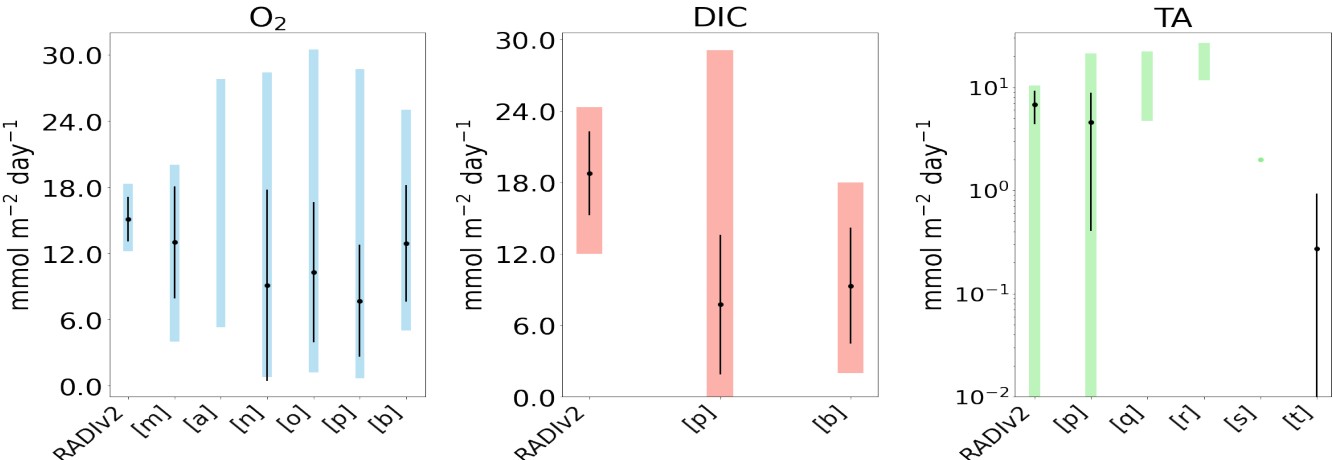

(a) Simulated benthic solute fluxes for the North Sea

(b) POC Remineralization and Percentage Remineralized

| POC remineralization rate [mmol C m$^{-2}$ d$^{-1}$] | |
| --- | --- |
| This study | 14.15–29.56 |
| North Sea[a] | 8.76–12.59 |
| North Sea[b] | 4–13.5 |
| Monterey Bay[c] | 8 |
| Washington Shelf[d; e; f] | 20 |
| Denmark shelf[g] | 16 |
| N. Carolina shelf[h] | 7–15 |
| Alaskan shelf[i] | 9–20 |
| New York shelf[j] | 11 |
| Maine shelf[k] | 11 |
| San Pedro shelf[l] | 23 |
| **Percentage Remineralized [%]** | |
| Aerobic, this study | 51–67 |
| SO$_4$ reduction, this study | 26–45 |
| Aerobic[a] | 46.9–89.3 |
| SO$_4$ reduction[a] | 10.7–54.1 |
| Aerobic[b] | 57–90 |
| Anaerobic[b] | 14–39 |

**Figure 2.** Comparison of simulated benthic solute fluxes and POC remineralization rates with values reported in the literature. (a) Simulated RADIv2 benthic solute fluxes for the North Sea compared to published values. Bars represent the range of values, while black dots with error bars indicate the mean and standard deviation, respectively. (b) POC remineralization rates and percentage of remineralization in different shelf environments. [a]Upton et al. (1993); [b]De Borger et al. (2021); [c]Berelson et al. (2003); [d]Archer and Devol (1992); [e]Christensen et al. (1984); [f]Christensen et al. (1987); [g]Canfield et al. (1993); [h]Anderson et al. (1994); [i]Grebmeier and McRoy (1989); [j]Rowe et al. (1988); [k]Christensen (1989); [l]Berelson et al. (2002); [m]Van Raaphorst et al. (1990); [n]Lohse et al. (1996); [o]McGinnis et al. (2014); [p]Brenner et al. (2016); [q]Burt et al. (2014); [r]Voynova et al. (2019); [s]Pätsch et al. (2018); [t]Van Dam et al. (2022).




## 3.2 Monterey Bay

RADIv2 simulations for Monterey Bay show similar trends to those for the North Sea, capturing the mean and standard deviation of benthic solute fluxes while underestimating their full observed range (Figure 3a). This discrepancy arises from the steady-state nature of the simulations, which do not account for short-term temporal variations, such as internal tidal waves (see North Sea subsection), which have been shown to significantly influence sediment dynamics in Monterey Bay (Cheriton et al., 2014). RADIv2 simulates an organic matter remineralization rate for Monterey Bay that is approximately 2–3 times higher than observed rates (Figure 3b). However, the values in Berelson et al. (2003) are lower than the global-mean $C_{org}$ remineralization rate for continental shelf environments ($\sim$12 mmol C m$^{-2}$ d$^{-1}$). The relative contribution of aerobic and anaerobic organic matter pathways simulated by RADIv2 is consistent with values observed in other continental shelf studies (Figure 3b) (Jørgensen, 1982) .

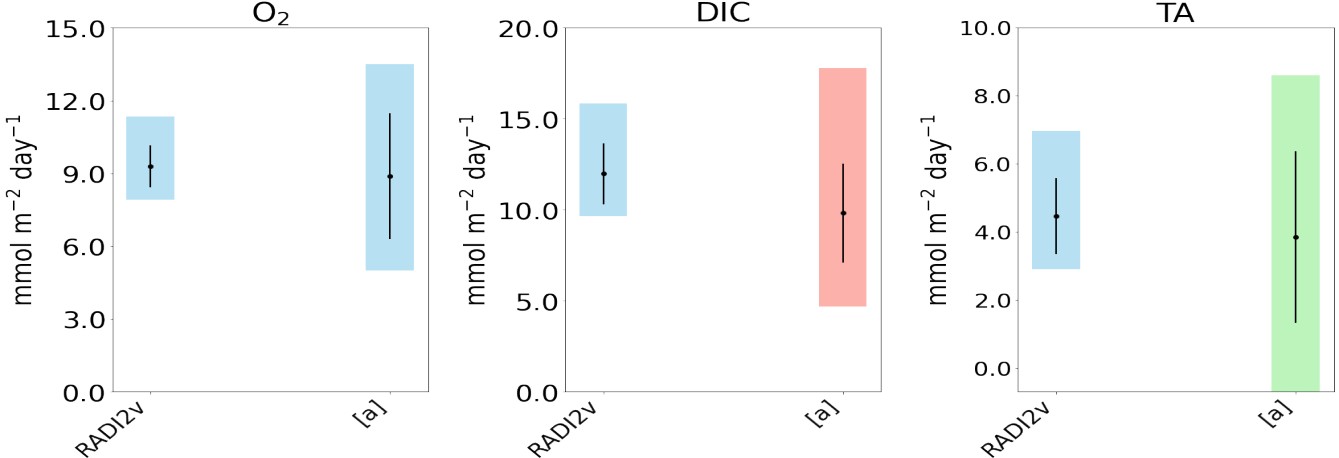

(a) Simulated benthic solute fluxes for Monterey Bay

(b) POC Remineralization and Percentage Remineralized

| POC remineralization rate [mmol C m$^{-2}$ d$^{-1}$] | |
|---|---|
| This study | 14.85–22.19 |
| Monterey Bay [a] | 8 |
| **Percentage Remineralized [%]** | |
| Aerobic, this study | 52–67 |
| SO$_4$ reduction, this study | 21–41 |

**Figure 3.** Comparison of simulated benthic solute fluxes and POC remineralization rates with values reported in the literature. (a) Simulated RADIv2 benthic solute fluxes for Monterey Bay compared to published values. Bars represent the range of values, while black dots with error bars indicate the mean and standard deviation, respectively. (b) POC remineralization rates and percentage remineralized values for this study compared to Monterey Bay in-situ data Berelson et al. (2003). [a]Berelson et al. (2003)



### 3.3 Iberian Margin

The Iberian Margin validation uses data from five transects spanning a wide range of depths, as reported by Epping et al. (2002). Two input files were created: one for shallow locations (< 400 m) and one for deeper locations (> 400 m). For the simulations, the average organic carbon decay rate $k_{avg} = p_f \times k_f + p_s \times k_{slow}$ is constrained between 1.5 and 2.25 a$^{-1}$ (Table

2), based on high-resolution in-situ oxygen microprofiles (Epping et al., 2002).

RADIv2 simulates oxygen fluxes that match the range of measured values (Figure 4a). However, the model tends to overestimate the $C_{org}$ remineralization rates (Figure 4b). The measured rates on the Iberian Margin are notably low when compared to other upwelling areas; shelf sediments off Peru exhibit rates of 5.3 ± 2.9 mmol C m$^{-2}$ d$^{-1}$ (Fossing, 1990), and margin sediments off Chile, up to depths of 2000 m, show rates approximately 15 times higher (Thamdrup and Canfield, 1996). The

observed rates in the Iberian Margin are lower than expected given the high primary production (∼7–11 mmol C m$^{-2}$ d$^{-1}$) in this region (Epping et al., 2002) and fall well below predicted values based on empirical relationships described in Middelburg et al. (1997).

Although DIC fluxes were not directly measured in this study, they can be inferred from the modeled O$_2$ influx using the Respiratory Quotient (RQ), defined as the molar ratio of DIC production to O$_2$ consumption during organic matter reminer-

alization (Jørgensen et al., 2022), $RQ = \frac{\Delta DIC}{\Delta O_2}$. This approach provides a practical means of evaluating DIC fluxes in the absence of direct in-situ measurements. Under idealized conditions, where organic matter is represented by redfield organic matter, CH$_2$O, the RQ equals 1.0, reflecting a 1:1 stoichiometric ratio between O$_2$ consumption and DIC production. However, re-evaluated marine phytoplankton composition lead to slightly lower RQ values, around 0.88–0.90 (Hedges et al., 2002; Anderson, 1995). Some studies also account for additional oxygen-consuming processes, particularly nitrification, which further

lowers RQ values (Tanioka and Matsumoto, 2020). Additional variability in RQ can come from incomplete reoxidation due to losses of N$_2$ or burial of FeS$_2$. FeS$_2$ can later be reoxidized through bioturbation-driven exposure to oxygen, further influencing oxygen demand and introducing additional variability in RQ in both directions. An empirical approach to deriving RQ values involves using in-situ measurements of benthic DIC efflux and O$_2$ influx. Flux-derived RQ values compiled by Jørgensen et al. (2022) span a broad range, from 0.69 to 1.31. The variation from theoretical stoichiometry can be attributed to factors includ-

ing carbonate dissolution, which enhances DIC fluxes without directly affecting O$_2$ consumption (Berelson et al., 1994), and the temporal decoupling of O$_2$ and DIC fluxes during episodic organic matter deposition (Therkildsen and Lomstein, 1993; Smith Jr et al., 2018). RADIv2 simulates average flux-derived RQ values around 1.0 for both shallow and deep sites on the Iberian Margin. These values are consistent with the Redfield stoichiometry assumed in the model and fall well within the empirical range reported by Jørgensen et al. (2022).




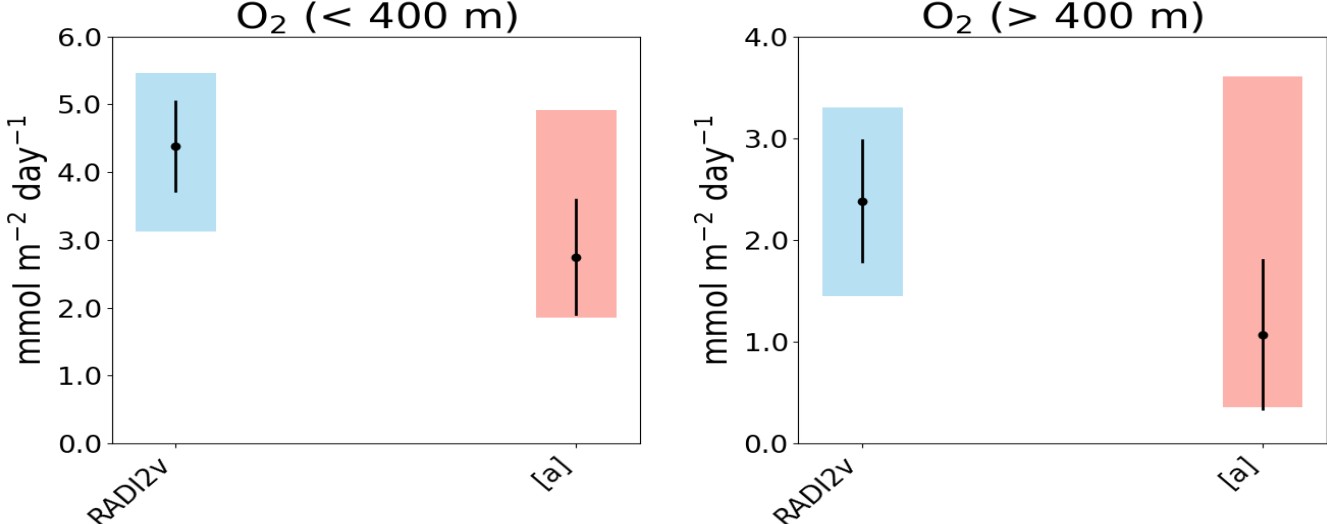

(a) Simulated benthic solute fluxes for Iberian Margin

(b) POC Remineralization and Percentage Remineralized

| POC remineralization rate [mmol C m⁻² d⁻¹] | |
|---|---|
| RADIv2 (< 400 m) | 3.99–6.93 |
| Iberian margin[a] (< 400 m) | 0.2–0.65 |
| RADIv2 (> 400 m) | 1.86–4.18 |
| Iberian margin[a] (> 400 m) | 0.04–0.45 |
| **Percentage Remineralized [%]** | |
| **< 400 m** | |
| RADIv2, aerobic | 54–93 |
| Iberian margin[a], aerobic | 37–88 |
| RADIv2, SO₄ reduction | 3–41 |
| Iberian margin[a], anaerobic | 10–60 |
| **> 400 m** | |
| RADIv2, aerobic | 69–97 |
| Iberian margin[a], aerobic | 30–98 |
| RADIv2, SO₄ reduction | 6–22 |
| Iberian margin[a], anaerobic | 0–65 |

**Figure 4.** Comparison of simulated benthic solute fluxes and POC remineralization rates with values reported in the literature. (a) Simulated RADIv2 benthic solute fluxes for the Iberian Margin compared to published values. Bars represent the range of values, while black dots with error bars indicate the mean and standard deviation, respectively. (b) POC remineralization rates and percentage remineralized values for this study compared to Epping et al. (2002). [a]Epping et al. (2002)





## 3.4 Arabian Sea

RADIv2 was validated using data from two deep-sea stations located in the Arabian Sea, WAST (4050-m depth) and SAST (4420-m depth), as described in Witte and Pfannkuche (2000). These stations were selected for validation due to their contrasting organic matter export fluxes, benthic carbon remineralization rates, and benthic oxygen fluxes.

WAST is characterized by high benthic $O_2$ uptake, one of the highest recorded in global deep-sea environments (Witte and Pfannkuche, 2000). This suggests that deep-sea sediments are not always stable, constant environments, but are influenced by variations in organic matter fluxes (Luff et al., 2000; Witte and Pfannkuche, 2000). The high benthic $O_2$ uptake is primarily driven by high particulate organic carbon (POC) rain rates of 1.38 $\mathrm{mol\,m^{-2}\,a^{-1}}$, resulting from monsoonal upwelling and lateral advection from the productive Arabian shelves Brink et al. (1998); Manghnani et al. (1998). These elevated POC fluxes lead to relatively high benthic remineralization rates compared to other deep-sea sites, although RADIv2 simulates higher values than observed (Figure 5b). Due to the higher POM rain, RADIv2 predicts aerobic benthic remineralization rates that align more closely with those observed in more shallow marine environments (Figure 5b). As a result, the simulated aerobic remineralization percentages are higher than those reported by Luff et al. (2000).

For SAST, located further from the shelf, the significantly lower POC rain rates lead to lower $O_2$ fluxes (Figure 5a), more in line with typical deep-sea $O_2$ uptake values (Witte and Pfannkuche, 2000). Here, RADIv2 overestimates both $O_2$ fluxes and remineralization rates, though the model's aerobic remineralization rates are consistent with those reported by Luff et al. (2000) (Figure 5b).

As no DIC measurements were available for these stations, we inferred DIC fluxes using the flux-derived RQ in RADIv2. The simulated RQ is 1.25 for WAST and 1.13 for SAST. Both are higher than those simulated for the Iberian Margin, due to more calcium carbonate dissolution. The values fall within the broader range of RQ values derived from in-situ flux measurements reported for marine sediments by Jørgensen et al. (2022).



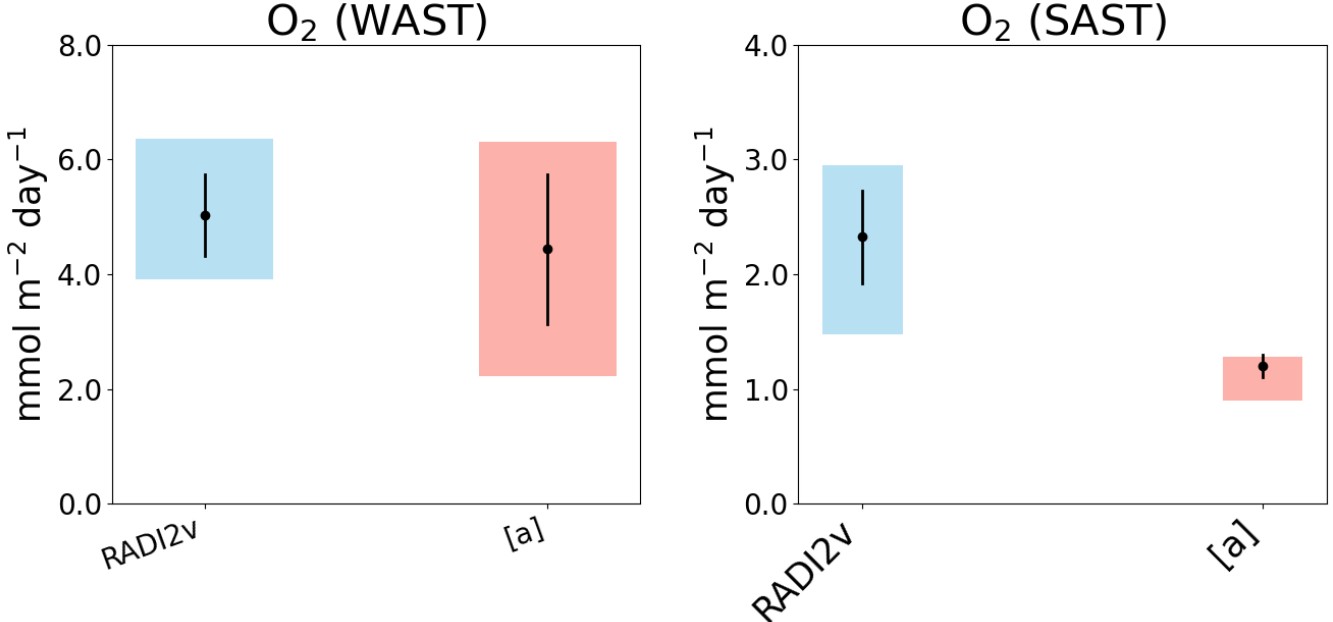

(a) Simulated benthic solute fluxes for Arabian Sea

(b) OM Remineralization and Percentage Remineralized

| OM remineralization rate [mmol C m⁻² d⁻¹] | |
|---|---|
| RADIv2 (WAST) | 6.5–10.7 |
| WAST[a] | 3.6 |
| RADIv2 (SAST) | 3.37–6.06 |
| SAST[a] | 0.6 |
| **Percentage Remineralized [%]** | |
| **WAST** | |
| RADIv2, aerobic | 53–74 |
| Aerobic[b] | 94 |
| **SAST** | |
| RADIv2, aerobic | 91–96 |
| Aerobic[b] | 97 |

**Figure 5.** Comparison of simulated benthic solute fluxes and OM remineralization rates with values reported in the literature. (a) Simulated RADIv2 benthic solute fluxes for the Arabian Sea compared to published values. Bars represent the range of values, while black dots with error bars indicate the mean and standard deviation, respectively. (b) OM remineralization rates and percentage remineralized values for this study compared to Witte and Pfannkuche (2000) and Luff et al. (2000). [a]Witte and Pfannkuche (2000); [b]Luff et al. (2000).



### 3.5 Summary of RADIv2 performance

The simulated ranges of solute fluxes in RADIv2 generally align with values found in the literature, although they are narrower. This narrowing likely results from the steady-state conditions employed in the simulations, which do not account for short-term variability, such as tidal and higher-frequency fluctuations. Despite this, the simulated mean fluxes match observations,

suggesting that RADIv2 provides a reliable representation of benthic solute fluxes over longer timescales. For locations without observed DIC fluxes, the empirical RQ, inferred from benthic DIC and $O_2$ fluxes, falls within the range of in-situ RQ values reported in the literature (Jørgensen et al., 2022). However, the model tends to overestimate organic carbon remineralization rates compared to observations.

### 3.6 Benthic-Pelagic Coupling in Ocean Models

Accurate representation of benthic-pelagic coupling is essential for simulating key biogeochemical cycles in both global and regional ocean models. However, full sediment-ocean coupling remains computationally expensive, limiting direct application in GOBMs. To overcome these challenges, we derive a regression-based universal metamodel that captures key solute SWI exchange while accounting for sediment heterogeneity across diverse marine environments. The metamodel incorporates bottom-water hydrodynamics and particulate export fluxes reaching the seafloor, parameterizing the corresponding SWI

exchange. This ensures a more accurate processing of particulates at the seafloor and accounts for the exchange of remineralization and dissolution products between sediments and bottom waters. By integrating these dynamics, our model addresses a limitation of current GOBMs, where benthic processes are either absent or represented using highly simplified parameterizations (Terhaar et al., 2024).

The metamodel represents benthic solute fluxes as linear functions of five key environmental drivers: bottom-water temper-

ature, current velocity, calcite saturation state, and POC and PIC deposition fluxes. These drivers are able to predominantly predict the benthic fluxes, and capture the variability in fluxes across diverse marine environments. By reducing the complex diagenetic processes simulated by RADIv2 into simplified mathematical relationships that accurately reproduce its outputs, the metamodel provides a robust and scalable representation of SWI exchange, making it well-suited for integration into large-scale ocean models.

The metamodel equations are as follows:

$$J_{\mathrm{DIC}} = 1.84 - 0.05 \cdot t - 1.68 \cdot U - 0.46 \cdot \Omega_{Ca} + 3.98 \cdot F_{POC} + 1.13 \cdot F_{PIC}, \tag{21}$$

$$J_{\mathrm{O_2}} = -1.00 + 0.0013 \cdot t - 4.80 \cdot U + 0.17 \cdot \Omega_{Ca} - 2.90 \cdot F_{POC} - 1.28 \cdot F_{PIC}, \tag{22}$$

$$J_{\mathrm{TA}} = 1.64 - 0.03 \cdot t - 8.52 \cdot U - 0.53 \cdot \Omega_{Ca} + 1.67 \cdot F_{POC} + 0.27 \cdot F_{PIC}, \tag{23}$$

where $J_{\mathrm{DIC}}$, $J_{\mathrm{O_2}}$ and $J_{\mathrm{TA}}$ are the DIC, $O_2$ and alkalinity sediment fluxes [$\mathrm{mmol\,m^{-2}\,day^{-1}}$], respectively. Positive fluxes

indicate solute release from sediments, while negative fluxes reflect sediment uptake. T is the bottom-water temperature [$^\circ$C],



U is the bottom-water current velocity [m/s], $\Omega_{Ca}$ is the bottom-water calcite saturation state [-], $F_{POC}$ is the POC rain rate [$\mathrm{mol\,m^{-2}\,a^{-1}}$] and $F_{PIC}$ is the PIC rain rate [$\mathrm{mol\,m^{-2}\,a^{-1}}$].

Figure 6 compares the simulated benthic solute fluxes for DIC, $O_2$, and TA from RADIv2 (x-axis) with the corresponding predictions from the metamodel (y-axis). The data points are color-coded by marine environment. The high $R^2$ values and
435 relatively low RMSE values demonstrate the metamodel's ability to accurately reproduce RADIv2 outputs, while also capturing flux variability across diverse marine settings.

**Figure 6.** Comparison of RADIv2 simulated benthic solute fluxes (x-axis) with metamodel predictions (y-axis) for DIC, $O_2$, and TA. Data points are color-coded by marine environment. Positive fluxes indicate solute release from sediments, while negative fluxes reflect sediment uptake.





## 4 Discussion and future development

### 4.1 RADIv2

In RADIv2, mass transport across the SWI occurs via diffusion or a combination of diffusion and dispersion. This is particularly
relevant in coastal sediments, where permeability is typically higher and bottom-water currents are stronger compared to deeper
regions, resulting in sediment fluxes that exceed diffusive fluxes alone (Jørgensen et al., 2022). In cases where sediments are
sufficiently permeable, a third transport regime, turbulent transport, can contribute to mass transfer across the SWI (Voermans
et al., 2018). The transitions between these three regimes (molecular diffusion, dispersive diffusion, and turbulent transport),
are characterized by the permeability Reynolds number, $Re_k = \sqrt{K}u_*/\nu$, where $K$ is the sediment permeability, $u_*$ is the
shear velocity and $\nu$ is the kinematic viscosity. $Re_k$ parameterizes the turbulent regime at the seafloor, with the permeability
of the sediments determining whether the turbulence can penetrate through the SWI. A low $Re_k$ indicates a regime dominated
by molecular diffusion, while a high $Re_k$ suggests that turbulent transport dominates mass transfer across the SWI.

These different regimes can be parameterized by effective diffusion coefficient (Voermans et al., 2018) and can be integrated
into future updates of RADIv2. However, the main limitation of this method is its reliance on sediment permeability, a variable
property that is challenging to measure in situ. While efforts are underway to map seabed permeability in coastal regions
(Moosdorf et al., 2024), a global dataset for linking permeability to RADIv2 is not yet available.

Future efforts will focus on improving the representation of organic carbon degradation kinetics, a key driver of sediment
fluxes. Currently, RADIv2 divides organic carbon into two reactivity classes ($k_{fast}$ and $k_{slow}$ ), each following first-order re-
action kinetics. This approach captures organic matter degradation in well-mixed sediments, where organic carbon reactivity
is relatively constant within the mixed layer (Kuderer and Middelburg, 2024). However, experimental studies (Westrich and
Berner, 1984), field observations (Jorgensen, 1978), and recent theoretical advances (Rothman, 2024) have shown that organic
carbon reactivity often declines continuously with degradation. In poorly mixed sediments or environments receiving highly
reactive organic carbon inputs, degradation rates vary significantly with sediment depth, and a simplified two-class model may
not accurately capture this variability. The work of Kuderer and Middelburg (2024) summarizes the conditions under which
different degradation kinetics are suitable, highlighting the need to model organic carbon reactivity as a continuum that de-
clines with age and burial depth in regions that receive highly reactive OM or poorly-mixed environments. Incorporating such
an approach into RADIv2 would enhance its ability to simulate organic matter degradation under a broader range of marine
sediments. Future work could focus on dynamically adjusting organic matter degradation kinetics based on the sediment mix-
ing regime (quantified by the Péclet number) and the reactivity of incoming organic carbon. This would also help address the
current overestimation of organic matter degradation rates by better capturing the declining reactivity of organic carbon over
time, providing more accurate remineralization estimates.

Both RADIv1 and RADIv2 are optimized for simulating carbon and oxygen cycling. However, future research could ex-
pand the model to include more detailed metal cycles, particularly for iron and manganese. For instance, incorporating more
comprehensive iron and manganese cycles, such as those described in Fossing et al. (2004), would enhance the model's rep-
resentation of metal dynamics in coastal environments. RADIv2 was designed with extensibility in mind, providing a flexible



and adaptable framework that makes the addition of new components or species straightforward. Moreover, RADIv2 can be updated to represent specific environments, such as mangrove forests, estuaries, or nearshore intertidal zones. This versatility enables RADIv2 to serve as a robust foundation for advancing future diagenetic research across a wide range of applications.

## 4.2 Metamodel

Significant discrepancies exist between estimates of the ocean carbon sink derived from $pCO_2$ products and those produced by ocean biogeochemical models (Friedlingstein et al., 2023; DeVries et al., 2023), which are further complicated by inter-model variability (Terhaar et al., 2024). One major uncertainty comes from the poor representation of particulate deposition at the seafloor and the resulting benthic fluxes from remineralization, dissolution, and exchange at the SWI. The exclusion of sediment-seawater exchange processes in current models limits their ability to accurately simulate coastal-ocean productivity

and air-sea gas exchange as benthic processes play an important role in nutrient recycling and carbon dynamics (Middelburg and Soetaert, 2004), particularly in shallow regions where the seafloor and bottom waters directly interact with the surface layer. An analysis by Séférian et al. (2020) of the marine biogeochemical components in CMIP5 and CMIP6 Earth System Models (ESMs) reveals that most CMIP5 models lacked any representation of sediments. While CMIP6 models have made significant progress incorporating sediment processes, their treatment remains highly simplified and, in some cases, is still omitted. When

sediments are not explicitly represented, particulates that reach the seafloor are either instantaneously remineralized (Swart et al., 2019) or permanently buried upon deposition (numerically lost) (Aumont and Bopp, 2006; Carroll et al., 2020). Most ESMs that include sediments rely on a box-model approach, where organic matter reaching the seafloor is either remineralized or buried based on simplified relationships with downward organic fluxes. Some ESMs estimate organic matter remineralization through denitrification following the formulation of Middelburg et al. (1996) (Stock et al., 2020; Aumont et al., 2015), with

additional pathways for aerobic respiration or sulfate reduction (Stock et al., 2020). More advanced sediment models, such as those incorporated in HAMOCC and PISCES-v2 (Ilyina et al., 2013; Mauritsen et al., 2019), adopt a layered sediment module based on Heinze et al. (1999). While this represents a significant step forwards, these models still rely on highly simplified treatments of key diagenetic processes. Important mechanisms, such as bioturbation (irrigation), dispersion, and anaerobic organic matter degradation are either absent or not fully accounted for.

Since RADIv2 is developed in Julia, while ESMs are typically written in FORTRAN, direct coupling is not straightforward without significant code translation. Additionally, fully kinetic diagenetic models like RADIv2 are computationally intensive, making their real-time integration into Earth System Models currently unfeasible. This study introduces a process-based meta-model that captures key diagenetic processes, often oversimplified or absent in existing parameterizations, without requiring full integration. The metamodel is tuned and validated for diverse marine environments, accurately reproducing SWI exchanges

simulated by RADIv2. By accounting for the heterogeneity and variability in sediment properties and environmental conditions, the metamodel is designed for broad applicability. It provides a more robust parameterization for solute benthic-pelagic coupling than current sediment representations used in GOBMs. The flexibility of RADIv2 allows for customization to specific marine environments, such as bays and estuaries, when region-specific data is available, enabling the generation of specialized metamodels for regional applications. Future work will focus on expanding the metamodel to include additional particulate




fluxes, such as particulate metallic fluxes, and on parameterizing the corresponding SWI exchange. Ongoing developments will also expand the range of solute fluxes covered by the metamodel, including phosphate, nitrate, ammonium, methane, and particulate organic and inorganic carbon fluxes. This expansion is expected to improve biogeochemical budget estimates, as the lack of sediment regeneration of nutrients from remineralized organic matter can lead to underestimation of primary production in biogeochemical models (Yool et al., 2013; Liu et al., 2007). The adaptability of the RADIv2 framework allows the approach

presented here to be extended to develop metamodels for any solute or particulate flux included in RADIv2, facilitating its application across diverse marine environments and geochemical processes.

## 5   Conclusions

RADIv2 is a versatile diagenetic model designed to simulate sediment-water interface (SWI) exchanges across a range of marine environments. Built using Julia's DifferentialEquations.jl package, RADIv2 leverages high-performance capabilities and

an adaptive structure to address the complexity and variability of sediment processes. A key feature of RADIv2 is the flexibility of its framework, which allows for adaptation to specific environmental conditions and simulation setups. This capability enables optimization for diverse marine environments, from dynamic coastal regions to stable deep-sea sediments. The ensemble simulation feature in RADIv2 constrains variable environmental parameters, such as bottom-water temperature, currents, and particulate export fluxes, by generating statistical envelopes that capture variability, better reflecting the natural fluctuations

observed in in situ measurements over time and helping to reduce uncertainty in model inputs. This approach enhances the model's ability to represent the heterogeneity of marine sediments. RADIv2 also includes an optimization scheme for tuning model parameters based on in-situ observations, further improving model accuracy. Simulated solute fluxes in RADIv2 across various marine environments generally align with observed values from the literature, though the narrower variability reflects the model's steady-state assumptions. Despite this, mean fluxes and empirically derived RQ values remain consistent with

in-situ data, indicating that RADIv2 provides robust estimates of benthic fluxes over longer timescales. However, the model tends to overestimate organic carbon remineralization rates.

     Using RADIv2 outputs, we developed a metamodel that parameterizes benthic-pelagic coupling with a simplified set of equations. These equations accurately capture steady-state benthic solute fluxes linked to the marine carbon budget across various environments, accounting for sediment heterogeneity and environmental variability. The metamodel builds on a com-

prehensive diagenetic framework that incorporates detailed sedimentary processes, without the high computational demands of fully coupled water column-sediment biogeochemical models at the global scale. This ensures more accurate processing of particulates at the seafloor and accounts for the exchange of remineralization and dissolution products between sediments and bottom waters. Ultimately, this metamodel offers a pathway for more accurate representations of benthic-pelagic coupling in global ocean biogeochemistry models, improving estimates of ocean biogeochemical cycling.



*Code and data availability.* The current version of RADIv2, implemented in Julia, is freely available from GitHub at https://github.com/ RADI-model (last access: May 2025) under the GNU General Public License v3.0. The exact version of the model used in this study, along with all input data and scripts required to reproduce the simulations and analyses presented in the paper, is archived on Zenodo (doi:10.5281/zenodo.15397043; (van der Zant et al., 2025)). RADIv2 users should cite both this publication and the corresponding Zenodo repository.

*Author contributions.* HFZ was responsible for conceptualization, methodology, software, validation, formal analysis, investigation, writing of the original draft, and visualization. OS was responsible for conceptualization, methodology, formal analysis, reviewing the manuscript and supervision of the project. JJM was responsible for the methodology, validation, and reviewing the manuscript. MPH was responsible for validation, software, and reviewing the manuscript. RS was responsible for the validation and reviewing the manuscript. DC was responsible for validation and reviewing the manuscript. DM was responsible for validation and reviewing the manuscript. KS was responsible for
validation and reviewing the manuscript. VLF was responsible for conceptualization, methodology, formal analysis, reviewing the manuscript and supervision of the project.

*Competing interests.* The contact author has declared that neither they nor their co-authors have any competing interests

*Acknowledgements.* This work is supported by an ERC grant Deep-C 101162777 to OS. DC, DM, and RS acknowledge support from the NASA Carbon Monitoring System program. As part of the La Rochelle Territoire Zéro Carbone (LRTZC) initiative, HFZ and VLF received
support from funding provided by the Plan d'Investissement d'Avenir (PIA) through the Territoires d'Innovation de Grande Envergure (TIGA), managed by the Caisse des Dépôts, as well as from the Région Nouvelle-Aquitaine (grant agreement 25875420).

We also acknowledge the use of artificial intelligence tools in the development of this work. GitHub Copilot (https://github.com/features/copilot) was used to assist in reviewing, optimizing, and improving the code for RADIv2.



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
