# Peer review of "RADIv2, an Adaptable and Versatile Diagenetic Model for Coastal and Open-Ocean Sediments"

_EGUsphere, 2025_

## Author Comment (AC1)

**AC:** We would like to thank both reviewers for their comments. Their suggestions have been extremely helpful in clarifying the scope of the work, strengthening the model evaluation, and improving the presentation of the manuscript. We have considered and implemented their recommendations where possible.

**RC1: Overview**

Van der Zant et al. present RADIv2, an updated version of the RADIv1 1D early diagenesis sediment model described by Sulpis et al., (2022). Specifically, they have included improved parameterizations for diffusive boundary layer thickness and pore water dispersion, and added methane cycling. They also added features that allow ensembles to be run, and that allow unknown parameters to be optimized to better fit in situ data. They validate RADIv2 from in situ measurements across a range of different locations, then use RADIv2 outputs to develop a metamodel. This metamodel parameterizes dissolved oxygen, dissolved inorganic carbon, and total alkalinity fluxes in terms of temperature, bottom velocity, bottom-water calcite saturation state, and organic and inorganic carbon fluxes. They suggest that this metamodel could be used to represent more complex sediment dynamics than are currently included in global ocean biogeochemical models (GOBMs), without incurring the significant computational expense of including a fully process based model.

**RC1: General comments**

**RC1: I think this paper adds some very useful features to RADIv1 and seeks to address a critical issue by demonstrating a pathway towards including more realistic sediment biogeochemical cycling in GOBMs. It's well written, and I think the authors have done a great job in finding so many datasets to compare RADIv2 output against as I know that useful datasets for calibrating such a model are not readily available. Overall, I think it's a strong paper. Below are some suggestions primarily focused on validation and the associated discussion which could improve it further.**

**RC1. I think this paper could be strengthened by adding more discussion to the sections validating RADIv2 against in situ observations. The applicability of RADIv2 and the resulting metamodel to a wide range of environments depends on these sections, hence I think they are particularly important.**

**AC:** We thank the reviewer for this suggestion and agree that the validation sections are central to demonstrating the applicability of RADIv2 and the metamodel. In the revised manuscript, we (i) added more comprehensive figures comparing modelled and observed benthic fluxes (Figures 2-5), (ii) introduced a statistical analysis that quantifies whether the model shows a significant mean bias relative to the observations (via a Welch two-sample t-test) and whether the simulated fluxes fall within the bulk (25–75 %) and overall (5–95 %) observational ranges, and (iii) expanded the site-specific discussion of these results in each validation subsection. In addition,

we included a sensitivity test for the remineralization rates simulated by RADIv2. These changes are described in more detail in our responses to the specific comments below.

**RC1: Comparisons between the modelled and observed fluxes currently focus on whether the modelled fluxes lie within the range of observational values. However, it seems like in several cases the model means lie the observed values. Performing some kind of statistical test to determine whether the model differs significantly from the observations or not would strengthen the case that the model does capture most of what's happening in the environment, and that the variations attributed to transient processes aren't as important. If they do differ significantly, it might indicate that these differences are due to more than just these transient variations, or that these variations occur so frequently that they need to be accounted for.**

**AC:** We agree with the reviewer that the comparison between modelled and observed fluxes can be strengthened.

For most of the literature datasets we use, we do not have access to the raw time series. The published values are typically site-mean fluxes and are presented with an associated standard deviation. For these locations, we do have the underlying distributions, and we cannot construct true violin plots or perform statistical tests at the level of individual measurements without generating pseudo-data and making additional assumptions about the underlying variability.

However, to better show how the model relates to the observations, we have revised Figures 2-5 that show model–data flux comparisons. Instead of simple range bars, we now show boxplots of the site-wise fluxes (25–75 % range) with median and mean lines, whiskers extending to the 5–95 % range, and overlaid points for each observational site. Where multiple datasets are available, points are colour-coded by source, using the published site means. This representation shows where the bulk of the observational fluxes lie, how different datasets contribute to the overall spread, and how the RADIv2 distribution overlaps with (or deviates from) this cloud of observations.

We added a simple statistical analysis to quantify whether the model exhibits a systematic bias relative to the observations. Because the observations and simulated values are not paired (they do not correspond to the same exact locations or boundary conditions) and the sample sizes differ, we use a Welch two-sample t-test to test for a difference in the mean flux between model and observations.

To account for the strong spatial heterogeneity of sediments, we also compare the spread of observed and modelled values. We define the "bulk" of the observations as the 25–75 % quantile range and the extremes as the 5–95 % quantiles. We then compute the fraction of modelled fluxes that fall within the observational bulk and the fraction that fall within the full 5–95 % observational envelope. Together, these metrics indicate whether the model is biased in its mean (systematically over- or underestimates the observed fluxes) and whether the simulated values

remain within the range documented in the literature. We have added the corresponding statistical results to the site-specific validation subsections (North Sea, Monterey Bay, Iberian Margin, and Arabian Sea) in the revised manuscript. For the SAST and WAST stations the number of observations is too small (n = 2 and n = 3, respectively) to apply these tests, so we rely there on a visual comparison only.

We believe this helps addresses the reviewer's comment about understanding whether differences between model and observations are systematic and if the long tails in the in situ data caused by few extreme points.

A description of the statistical analysis added in **L300** of the manuscript: To quantitatively assess how well RADIv2 represents this coupling, we compare the distributions of modelled and observed benthic fluxes where sufficient observational data are available. Because observations and simulations are not always paired in space or forcing (not all parameters or boundary conditions are available to fully constrain RADIv2) and sample sizes differ, we use a Welch two-sample t-test to evaluate whether the mean modelled flux differs significantly from the mean of the observations. To account for the strong spatial heterogeneity of benthic fluxes, we also quantify the fraction of simulated values that fall within the observational interquartile range (25–75 %) and within the broader 5–95 % observational envelope. Together, these diagnostics provide a joint measure of systematic bias (over- or underestimation) and of how well the modelled fluxes remain within the range documented in the literature.

**RC1: I also think the discussion about the discrepancies between the model output and the observational data could dive deeper into factors that might be affecting these differences on the modelling side. Currently much of this discussion is framed around how the values the model is being compared to are outliers in the wider literature (e.g., the North Sea values for remineralization rates were lower than rates reported from other shelves, the Berelson et al., (2003) values for Monterey Bay are lower than the global-mean $C_{org}$ remineralization rates for continental shelf environments). I think that if the point of this section is to demonstrate that RADIv2 can capture conditions across a range of different environments then it's important to focus the discussion on how it compares to the specific environment it's being tested against rather than against more generic conditions.**

**AC:** We agree with the reviewer that the discussion should focus more on the specific environments used for validation rather than comparing to global "typical" conditions. In the revised manuscript we have rewritten the site-specific validation sections to discuss factors that likely drive discrepancy. More detail and a sensitivity analysis are also added for the remineralization rates, as suggested by the next comment in section 3.5 (Summary of RADIv2 performance).

In the revised North Sea subsection, we now state explicitly that the simulated DIC fluxes and remineralization rates tend to be higher than the local observations, and that this likely reflects an

imposed organic-matter supply and/or reactivity that is too high. In our set-up, a highly reactive organic-matter pool is supplied to the sediment, so most of the degradation occurs close to the sediment–water interface. This may enhance DIC fluxes (and, but to lesser extent, elevate O2 uptake) compared to the measured values. The POM rain in Table 2 is estimated from mean yearly values described in Upton et al. (1993). Such annual means smooth over blooms and short-lived export pulses and may overestimate the effective POM supply at the time of the benthic flux measurements.

**L327**: The simulated DIC fluxes and remineralization rates tend to be higher than the literature range. This likely reflects a combination of too strong organic-matter supply and too reactive organic-matter parameters in our set-up. The imposed POM rain is based on annual-mean export estimates, which smooth over episodic blooms and short-lived pulses and thus likely overestimate the supply at the time of the benthic flux measurements. As a result, most degradation occurs near the sediment–water interface and drives larger DIC fluxes (and slightly elevated $O_2$ uptake) than those inferred from the local measurements and modelling studies.

And **L336**: The modeled remineralization rates (Figure 2b) are within the same order of magnitude than those observed in continental shelf environments, although they exceed observations from the North Sea. This positive bias is consistent with the combination of relatively high imposed organic-matter supply and reactive organic-matter parameters discussed above.

For Monterey Bay, we now state explicitly that the imposed POM flux is not measured in situ but estimated by analogy with the North Sea calibration, and that this choice likely overestimates the local organic-matter supply and contributes to the positive bias in remineralization and solute fluxes.

**L348**: Because Berelson et al. (2003) did not measure POM rain to the seafloor directly, the POM flux range used in Table 2 was estimated by analogy with the North Sea calibration. This choice likely overestimates the actual organic-matter supply at Monterey Bay and contributes to the positive bias in the simulated remineralization rates.

For the Iberian Margin, we highlight that the original study already noted the unexpectedly low remineralization rates given the high primary production and did not resolve their cause. We adopt the relatively high export fluxes reported in the paper without additional site-specific tuning, which likely causes RADIv2 to overestimates local remineralization rates at these stations.

 **L363**: In our set-up, we adopt the export flux estimates from Epping et al. (2002) without additional site-specific tuning, so uncertainties in the effective POC supply at these stations likely contribute to the positive bias in RADIv2 remineralization rates.

**RC1: Diving more into the factors in the model that could be driving these discrepancies could also offer more insight into ways that the RADIv2 could be improved. In particular, I think that more discussion of why RADIv2 overestimates the POC remineralization rate at each site would be useful, as it seems like these overestimations could be responsible for the modelled O2, TA, and DIC fluxes tending to be higher than the observed values. Could this indicate that the POC fluxes used for the model are too high? How are the remineralization rate constants chosen for each site, and how does changing them affect both the remineralization rates and the partitioning of remineralization between aerobic and anaerobic pathways? There are also other parameters that could affect these fluxes, such as changes in porosity or the temperature dependence of redox reaction rates. I understand that RADIv2 has a lot of dials to turn, and the data necessary to really constrain it doesn't exist and so these questions are probably impossible to answer definitively. However, given the importance of this section I think more discussion about why it overestimates remineraliton would help strengthen this work.**

**AC:** We agree that it is important to understand which model parameters are primarily responsible for the overestimation of organic matter remineralisation. In RADIv2, the total remineralisation rate is set by the OM reactivity parameters (k_fast, k_slow and their partitioning). The high remineralisation rates are thus a consequence of the assumed POC rain and two-G kinetics. For most sites the imposed POC rain rates are taken from literature estimates that are either annual/climatological means or derived from export-productivity relationships, rather than being co-located and time-aligned with the in-situ measurements.

In the revised manuscript we add a sensitivity experiment to illustrate their relative impact on the remineralizaiton rates.

RADIv2 uses uniform degradation rate constants and a fixed partitioning between the fast and slow pools.  To quantify how much of the disagreement could in principle be absorbed by either the POM flux or the reactivity, we performed three sensitivity tests at the Iberian Margin shallow calibration site, holding all other parameters fixed

FPOM sweep: we scaled the imposed POM rain by factors 0.5, 0.75, 1.0, 1.25 and 1.5, while keeping reactivity kinetics constant between runs. The resulting total remineralisation rates were:

2.54, 3.54, 4.52, 5.50 and 6.46 mmol C m$^{-2}$ d$^{-1}$.

Changing FPOM by ±50% changes total remineralization by roughly −44% to +43%, and the response is close to linear.

Reactivity sweep: we scaled both k_fast and k_slow by factors 0.5, 0.75, 1.0, 1.5, 2.0, and kept their partitioning and the POM rain constant between runs.

The corresponding total remineralization rates were:

5.39, 4.84, 4.52, 4.34 and 4.39 mmol C m$^{-2}$ d$^{-1}$.

The response is non-monotonic: increasing the rate constants does not simply translate into proportionally higher integrated remineralization, because faster remineralization near the surface reduces the amount of POM that is transported to depth and can be degraded there.

Partitioning sweep: we scaled the fraction of fast POM by 0.5, 0.6, 0.7, 0.8, 0.9 (with the slow fraction equal to 1 – fast) and keep the reaction rates and POM rain constant.

The corresponding total remineralisation rates were:

4.31, 4.36, 4.40, 4.45, 4.49 mmol C m$^{-2}$ d$^{-1}$.

Small changes in the vertically integrated rate are simulated over the different partitionings. This knob mainly affects where in the column and which electron acceptors dominate remineralisation, rather than the total C oxidation at fixed FPOC and rate constants.

Taken together, these tests show that (i) uncertainties in the local POC supply are sufficient to produce significant changes in the total remineralisation rate, and (ii) global rescaling of the organic matter kinetics and their fast/slow partitioning also affects the integrated rate, but to a lesser degree and in a more non-linear fashion. It should be noted that, FPOC, k_fast/k_slow and their partitioning are not independently constrained at most sites, and different combinations of these parameters can yield similar total remineralisation rates. At the same time, the sensitivity tests suggest that RADIv2 is not locked into a "too-reactive" regime. The systematic positive bias in some of the case studies is consistent with uncertainties in the local POM supply, and/or to lesser extent, the effective reactivity of the sedimentary organic matter.

We have added a description of this sensitivity test and its implications to the revised manuscript (Section 3.5 Summary of RADIv2 performance) starting in **L408**: However, the model tends to overestimate organic carbon remineralization rates compared to observations. For most sites, the POM rain rates are taken from literature estimates or export–production relationships and are not co-located and time-aligned with the individual in situ flux measurements, so uncertainties in the local POM supply and its degradation kinetics are likely the source of this positive bias. To quantify how much of the disagreement could in principle be absorbed by either the POM flux or the reactivity, we performed three sensitivity tests at the Iberian Margin shallow calibration site, holding all other parameters fixed.  Scaling the imposed POM rain by factors 0.5–1.5, while keeping all other organic-matter parameters constant, changes the vertically integrated remineralization rate by roughly -40 to +40 %, with an almost linear response. In contrast, globally scaling both k_fast and k_slow by factors over a range of 0.5–2.0, while keeping the fast/slow partitioning and POM rain constant, produces smaller and more non-monotonic changes in the integrated rate because faster degradation near the surface reduces the amount of POM transported to depth. Varying the partitioning between the fast and slow pools (while keeping total POM rain and rate constants fixed) only weakly affects the depth-integrated

remineralization, but does shift where in the sediment column carbon is oxidized and hence the balance between aerobic and anaerobic pathways. It should be noted that, FPOM, k_fast/k_slow and their partitioning are not independently constrained, and different combinations of these parameters can yield similar total remineralisation rates. These tests indicate that uncertainties in the local POM flux are sufficient to account for much of the overestimation of remineralization, and that global rescaling of the organic matter kinetics and their fast/slow partitioning also affects the integrated rate, but to a lesser degree and in a more non-linear fashion. At the same time, the sensitivity tests suggest that RADIv2 is not locked into a "too-reactive" regime. The systematic positive bias across case studies is consistent with uncertainties in the local POM supply, and/or to a lesser extent, the effective reactivity of the sedimentary organic matter.

**RC1: I think it would also be useful to check the metamodel against the experimental data, rather than only comparing it to RADIv2 output. The ultimate goal of the metamodel is to simulate the world, hence I think it's important to see how it does against observational data in addition to how well it replicates the results of RADIv2.**

**AC:** We agree that as the metamodel is to be implemted in GOBMs, the metamodel should be evaluated not only against RADIv2, but also against in-situ data.

Our primary design goal for the metamodel is to act as a surrogate for RADIv2 inside global ocean models: it is trained exclusively on RADIv2 output and is not calibrated directly to observations. Any mismatch with in-situ data can be decomposed into (i) structural/model error in RADIv2 itself and (ii) an emulation error introduced by the metamodel (i.e. the fact that the metamodel–RADIv2 fit is not perfect, $R^2 < 1$ and RMSE $> 0$).

To better understand this emulation error, we performed an additional offline test at the North Sea sites. We forced the metamodel with the same physical and biogeochemical parameters (T, FPOC, FPIC, $\Omega\_Ca$, and bottom-current speed U) as in the corresponding RADIv2 set-up and diagnosed the same benthic fluxes ($O_2$, DIC, TA). We then compared both RADIv2 and metamodel predictions to the in-situ flux data.

A full global-scale assessment of metamodel skill within a coupled GOBM is beyond the scope of this paper. This is mainly because the simulated fluxes by the metamodel inside of a GOBM would also reflect biases of the host model itself.

We have added a description of this experiment and the Figure in the revised manuscript starting at **L437:** We performed an offline comparison of the metamodel against in situ benthic flux measurements at the North Sea sites. The metamodel is explicitly designed as a surrogate for RADIv2: its coefficients are calibrated on RADIv2 ensemble outputs and are not tuned directly to observations. Any mismatch with in situ data can therefore be separated into biases and structural limitations in RADIv2 itself, and an additional emulation error introduced by the regression ($R^2 < 1$ and RMSE $> 0$). To quantify this emulation error and to compare the metamodel directly to both observations and RADIv2, we ran the metamodel with the same

bottom-water and depositional drivers used in the corresponding RADIv2 North Sea ensemble set-up (Table 2). Figure 7 compares the resulting metamodel-predicted fluxes with the underlying RADIv2 fluxes and the in situ observations.

[Figure]

**RC1: The explanation of how the metamodel was developed could be made clearer by specifying which results the regressions were performed on. If they were performed on the same outputs used to validate the model, then I think it would be helpful for that to be stated. If that is the case, was there thought given to running RADIv2 over a wider parameter space and performing regressions on those results? One area that could potentially be beneficial to explore would be more tropical, carbonate rich environments. Assessing the performance of a metamodel developed over a wider parameter space against both RADIv2 outputs and the validation sites presented here would provide a better basis for assessing how well the metamodel would fare over the diverse range of environments present within a GOBM.**

**AC:** We agree that the description of the metamodel training should be more explicit. The current metamodel is indeed trained on the same set of RADIv2 experiments that we use for the flux comparison figures. For each site, we constructed an ensemble that spans the observed variability in the key drivers (bottom-water temperature T, current speed U, organic carbon flux FPOC, inorganic carbon flux FPIC, and bottom-water $\Omega$Ca), and we then fit the linear metamodel to the resulting RADIv2 fluxes. A practical challenge was to identify sites that provide sufficient data to calibrate and validate RADIv2, while still sampling a range of environmental conditions without over-representing particular settings (e.g. a few very well-studied regions that represent only a small fraction of the ocean) in the regression. We did consider training the metamodel over a much wider parameter space, however, without a consistent carbonate-chemistry inputs and site-specific constraints, very broad ranges in POC,

FPIC and ΩCa can generate physically unrealistic combinations (for example, low forced calcite saturations states with, very high PIC fluxes and low pressure). As all simulations performed are the basis of the metamodel, RADIv2 has not yet been tested in such regions, and the metamodel will lack implicit representation of these settings (as is also the case for deltas and very shallow regions, noted by Reviewer 2). We agree that extending the training domain towards more different environments is a logical next step and would improve the ability of the metamodel to perform better in these regions.

In the revised Discussion we now explicitly state that the present metamodel lacks representation of environments such as carbonate-rich tropical shelves. We also highlight that training on a wider, yet physically consistent, ensemble of simulations is a key avenue for future improvement, especially for more localized applications.

The explanation of how the metamodel was developed is now more explicitly stated in **L322:** The same ensemble of RADIv2 fluxes forms the basis of the metamodel introduced in the next section.

And in **L413:** The metamodel is calibrated by fitting it to RADIv2 benthic fluxes from the ensemble simulations summarized in Table 2 and Figures 2–5, i.e., the same experiments used to evaluate RADIv2 against observations.

The limitations of the trained dataset for the metamodel are now added to the discussion starting in **L502**: However, the present training and validation set primarily samples temperate shelves, margins, and deep-sea settings, and does not explicitly include environments such as deltaic systems, very shallow nearshore regions, or carbonate-rich tropical shelves. Expanding the training ensemble to a wider, yet physically consistent, range of bottom-water conditions and particulate fluxes, that can also be validated with in-situ observations, represents an important future improvement, especially for more localized applications.

**RC1: I think the structure of the paper could be changed slightly to improve clarity. Section 3.1 to 3.5 are all about model validation, so I would suggest renaming section 3 to model validation. I would then add a distinct metamodel section, which could include parts of what is written in section 3, and 3.6**

**AC:** We agree with the reviewer that this improves the clarity of the paper. Section 3.1 to 3.5 have been renamed to as model validation and a distinct metamodel section (section 4) is added for what was previously section 3.6. Moreover, in the revised manuscript, sections 3.1 and 4 has been adapted, such that sections 3.1 now more refer to the the validation of RADIv2 against in-situ measurements, without emphasis on the metamodel. The metamodel introduction and explanation has been moved to section 4.

**RC1: Overall, I think this is a strong piece of work that highlights a path forward and makes valuable contributions towards addressing a large problem in current GOBMs. The**

**suggestions above are intended to help strengthen the validation and discussion in order to further improve the work and maximize its impact. I congratulate the authors for tackling this difficult problem, and look forward to seeing this work develop further.**

**Specific comments**

**RC1: Line 59: generic is repeated twice**

**AC:** Thank you, repetition is removed.

**RC1: Line 107: How is this incorporated within RADIv2? Manually setting them based on literature values?**

**AC:** The reactivity of the OM pools can be set manually, for example based on literature values or measurements, or it can be calculated from the empirical flux relationship used in RADIv1 (tuned for deep-sea settings). We now have more explicitly stated this starting from **L107:** The rate constants for the fast and slow organic carbon pools can be specified directly by the user, for example based on literature values or in-situ measurements, or they can be diagnosed from the empirical flux relationship used in RADIv1 (tuned for deep-sea settings (Sulpis et al., 2022)).

**RC1: Line 112: Does RADIv2 use the same carbonate dissolution/precipitation scheme as RADIv1? If so, I think it's worth stating here, as RADIv1 is setup for relatively cold temperatures. If not, I think it would be useful to describe the differences given the importance of precipitation and dissolution to sediment biogeochemistry.**

**AC:** RADIv2 uses the same carbonate dissolution/precipitation scheme as RADIv1. In both versions, calcite dissolution is represented with rate laws based on the Naviaux et al. (2019) experiments, which were carried out at 5 °C in seawater and provide what we consider to be the most comprehensive set of kinetic constraints currently available. However, we agree that the underlying empirical laws are best constrained for cold conditions, and do not explicitly resolve all possible temperature and undersaturated Ω regimes (e.g. the transition between step retreat, defect-assisted etch pits and homogeneous etch pit formation at higher temperatures and Ω < 1, as discussed by Naviaux et al. 2019).

We have clarified this in the revised manuscript by explicitly stating that RADIv2 inherits the RADIv1 carbonate dissolution/precipitation scheme in **L112:** RADIv2 uses the same carbonate dissolution–precipitation scheme as RADIv1, non-linear kinetics for both calcite and aragonite dissolution described by Naviaux et al. (2019) and Dong et al (2019).

**RC1: Line 256: Move the first bracket to after ITTC et al., so that it reads … ITTC et al., (2011).**

**AC:** Thank you for noticing this. We have changed the in-text citation accordingly.

**RC1: Line 296: It could be helpful to be a bit more specific here. Is it the climate change-induced warming itself that could outpace the capacity, or is it the effects of warming? If so, what effects specifically?**

**AC:** Our intention was to refer to the effects of warming on methane cycling, rather than to temperature alone. Warming can enhance methane production and upward transport in sediments, whereas the anaerobic oxidation of methane (AOM) is not found to become more efficient with increasing temperature. As a result, the warming-induced increase in methane supply can outpace the capacity of AOM to consume it, allowing a larger fraction of seafloor methane emissions to bypass the sedimentary filter.

We have now explained this more explicitly in **L296**: However, recent studies suggest that anaerobic oxidation of methane does not become substantially more efficient with warming, whereas warming can enhance methane production within the sediments (Stranne et al., (2022). As a result, a larger fraction of climate-driven seafloor methane production may bypass the sedimentary filter and reach the water column.

**RC1: Line 336:**

**AC:** Empty comment.

**RC1: Line 348: A citation would be handy for the 12 mmol C $m^{-2}$ $d^{-1}$ number.**

**AC:** We agree with the reviewer. This value is from Berelson et al. (2003) and added in the text.

**RC1: Line 392: The text refers to Luff et al. (2000) while the figure reports values from Epping et al. (2002). Should there be another reference in here?**

**AC:** Line 392 refers to the Arabian Sea stations shown in Fig. 5a–b, where the observations come from Witte and Pfannkuche (2000) and the subsequent modelling study by Luff et al. (2000). The Iberian Margin data from Epping et al. (2002) are used separately in Fig. 4, not in Fig. 5.

**RC1: Line 426: Capitalise the T variables.**

**AC:** Thank you for noticing this. We have capitalized the T variables in Eq. 21-23.

**RC1: Line 457: By keeping track of organic carbon pools with different labilities, won't RADIv2 capture some of this implicitly? The more labile stuff will be consumed closer to the surface, resulting in a reduction in lability with depth.**

**AC:** Yes, that is correct. RADIv2 does implicitly capture part of the depth structure in organic matter lability. The two-G formulation with a fast and a slow pool naturally leads to vertically structured degradation: the more labile pool is consumed near the sediment–water interface, and deeper layers have less reactive carbon. In that sense, the bulk effective reactivity of total POC

indeed decreases with depth. However, reactivity of each pool (k_fast, k_slow) is fixed, and does not change with depth or time. This means that ageing and progressive "internal" loss of reactivity within a given pool are not represented. Moreover, in heterogeneous or poorly mixed sediments, or in settings with highly reactive, pulsed inputs, Kuderer and Middelburg (2024) have shown that reactivity with depth is better described by continuous reactivity models (i.e. a spectrum of k-values or power-law decay), rather than a small number of fixed-k pools.

We have clarified this starting from **L454**: This approach to organic matter degradation produces a vertical structure in bulk (apparent) reactivity: the more labile pool is preferentially consumed near the sediment–water interface, so deeper layers are dominated by less reactive carbon. This is sufficient for well-mixed sediments, where organic carbon reactivity is relatively constant within the mixed layer (Kuderer and Middelburg, 2024). However, experimental studies (Westrich and Berner, 1984), field observations (Jorgensen, 1978), and recent theoretical advances (Rothman, 2024) have shown that organic carbon reactivity often declines continuously with degradation. In RADIv2 each organic-matter pool has a fixed degradation rate constant, and degradation does not weaken with depth or time. The work of Kuderer and Middelburg (2024) summarizes the conditions under which different degradation kinetics are suitable, and highlights the need to model organic carbon reactivity as a continuum that declines with age and burial depth in regions that receive highly reactive OM or poorly-mixed environments, rather than a small number of fixed-k pools. Incorporating such an approach into RADIv2 would enhance its ability to simulate organic matter degradation under a broader range of marine sediments.

**Figures & tables**

**RC1: Figure 1: The fitted line in 1a is difficult to make out. It might be worth changing the colour and style so it stands out a bit more. Switching the panels may also improve the flow so that they appear in the order that they're discussed.**

**AC:** Yes, we agree. Panels (a) and (b) have been swapped, in-text references to them changed accordingly and the fitted line has increased line width.

[Figure]

**RC1: Figure 2: Changing the x axis labels so they're in alphabetical order would make it easier to find the reference each bar refers to. You could also put all of the references in a single table, either in the text or supplementary, so that you didn't have to list them in each caption.**

**AC:** With the new Figures (2-5) that better show how the model relates to the observations, the citations are no longer on the x-axis but now in the legend of the Figures.

**RC1: If the data's available and n is large enough that they're not too noisy, it would be interesting to see the bars as violin plots to better get a sense of the data. Are the long tails in the in situ data caused by just a few extreme points, or do some of these data sets of some kind of bimodality? Are the RADIv2 results relatively normal?**

**AC:** This is a great suggestion, but for most of the literature datasets we use, we do not have access to the raw time series. The published values are typically site-mean fluxes with an associated standard deviation. As a consequence, we do not know the full shape of the underlying distributions, and we cannot construct "true" violin plots or perform hypothesis tests at the level of individual measurements without generating pseudo-data and making additional assumptions about the underlying variability and sampling design.

To improve the comparison and better convey how the model relates to the observations, we have revised the all Figures (2-5) that show model–data flux comparisons. Instead of simple range bars, we now plot a boxplot of the site-wise fluxes and overlay points for each observational site, colour-coded by dataset, using the published site means. This representation shows where the bulk of the observational fluxes lie, how different datasets contribute to the overall spread, and how the RADIv2 distribution overlaps with (or deviates from) this cloud of observations. Moreover, we have added the corresponding statistical results discussed in the previous response to the site-specific validation subsections (North Sea, Monterey Bay, Iberian Margin, and Arabian Sea) in the revised manuscript.

[Figure]

[Figure]

**RC1: Could the table be presented as a figure as well? Given the number of different values reported it might make it easier to quickly compare different values.**

**AC:** We explored several plotting options (grouped bars and point–range plots), but the strong heterogeneity between datasets (different numbers of sites and inconsistency in ranges and their uncertainties) made the graphical summaries hard to interpret. For this reason, we chose to retain the tabular format.

**RC1: Figures 3-5: Same comment about violin plots, though I think a table's fine for just a few values. The figure panels for 4 and 5 could potentially be narrowed. The colours in figure 2 correspond to the parameter being plotted. It would be good to keep these colours consistent in the other plots.**

**AC:** We agree with the reviewer that the comparison between modelled and observed fluxes can be strengthened. Given the limited number of site-mean values for these locations, we did not implement violin plots (see previous comment and response), but we now explicitly show all individual observational points in the panels, together with the model distributions.

**RC1: Figure 6: The x labels have RADI2v instead of RADIv2. All the other figures with oxygen fluxes (which I think were into the sediment) have reported them as positive values while here they're reported as negative values. I think changing the other figures to have negative values would help make the presentation of this paper more consistent, and highlight that the oxygen fluxes are in the opposite direction to the TA and DIC fluxes.**

**AC:** We thank the reviewer for pointing this out. The label has been corrected to "RADIv2" in the revised version. We agree that reporting $O_2$ fluxes as negative values is more clear, making the opposite direction of $O_2$ uptake (into the sediment) versus DIC and TA release explicit. We have therefore updated all figures to use a consistent sign convention, with $O_2$ fluxes shown as negative and DIC/TA fluxes as positive.

[Figure]

**RC1: Table 2: Is saturation state calculated from TA and DIC? If so, it might be worth stating it specifically given the larger uncertainties that can occur if you calculate it from other pairs of parameters.**

**AC:** In the revised manuscript we now explicitly state that ΩCa and ΩAr are computed from TA and DIC using a the CO2SYS carbonate chemistry routine and provide the Zenodo DOI for the CO2System.jl module starting in **L324:** For each site, the calcium carbonate state is computed from total alkalinity and dissolved inorganic carbon using the CO2SYS carbonate-chemistry routine (Humphreys et al., 2022), which solves the carbonate system with standard equilibrium constants and calcite/aragonite solubility products following Mucci (1983) and applies pressure

corrections following Ingle et al. (1973). The CO2System.jl module is available on https://doi.org/10.5281/zenodo.6395674.

**RC1: It seems like all the organic carbon is now either fast or slow, whereas RADIv1 had refractory organic carbon. Has that been removed from the model? If so, I think this change should be mentioned somewhere.**

**AC:** RADIv2 still allows for a refractory organic carbon pool. In the ensemble simulations presented in Table 2, we assume that the incoming organic matter flux is entirely partitioned into a "fast" and a "slow" reactive pool. Refractory organic carbon is included in the model structure, but its reactivity is fixed to zero, so it does not contribute to remineralization or benthic fluxes. This choice avoids ambiguity when applying the metamodel in larger-scale models: if the training were based on a total POC flux that included a refractory fraction, but the same fraction was not treated consistently in the host model, the predicted benthic fluxes would be systematically biased. Users who wish to track inert carbon inventories can still include a refractory fraction in RADIv2 applications.

We now mention that RADIv2 still includes a refractory carbon pool in **L104**: A refractory organic carbon pool is also retained in the model structure, but its reactivity is set to zero by default, so it does not remineralize. The refractory fraction can be prescribed as a user-defined fraction of the incoming POM rain, in which case it mainly affects sediment burial rates (computed from the total particulate rain to the sediments) and the bulk carbon content of the sediment column.

And further explain that only reactive OM is considered in the ensemble runs for the metamodel starting in **L324:** For the metamodel training ensembles, we set the refractory fraction of particulate organic matter to zero and treat all incoming POM as reactive (assigned to the fast or slow pools), because the POM flux, converted to POC according to a fixed stoichiometry, is used directly as a predictor. In RADIv2, refractory organic carbon does not contribute to benthic solute fluxes, so including a refractory fraction in the training data would introduce ambiguity when applying the metamodel in larger-scale models: if the training was based on a total POC flux that includes a refractory fraction, but the same fraction is not treated consistently in the host model, the predicted benthic flux

---

## Author Comment (AC2)

**AC:** We would like to thank both reviewers for their comments. Their suggestions have been extremely helpful in clarifying the scope of the work, strengthening the model evaluation, and improving the presentation of the manuscript. We have considered and implemented their recommendations where possible.

**RC2: Overall Comment**

**The paper by Van der Zant presented an update to the previously published Radi early diagenetic model. This version of the model improves upon first version by including relevant processes related to the exchange of solutes across the sediment-water interface (SWI) as well as enhancing the model capabilities for simulating biogeochemical processes in highly permeable sediments. A key feature of this model and the overall paper is the showcasing of the adaptability of the model in computing benthic solute fluxes which can be used in coupled benthic pelagic application from a metamodel built from RadiV2. This parameterization of benthic solutes exchange from metamodel is important in the context of better representation of seafloor biogeochemical dynamics in global earth system model. Overall, the paper is well written and concise. The paper take care in explaining the relevant addition of the new development of Radi and showcasing its utilizing in estimating benthic fluxes across different marine environment. However, the paper will do well in at least providing supplementary text on the performance of this new version in capturing the variability of porewater profiles especially in coastal regions where it has previously not been tested. This will help inform users of the strength of the new model as well as the current limitation as highlighted in the latter part of the paper for their application. Overall, this is solid work, and I recommend for publication following some minor comments below.**

**AC:** We thank the reviewer for this suggestion and agree that a porewater comparison in a coastal setting strengthens the evaluation of RADIv2.

In the supplementary data we have added a dedicated test case for the shallow Iberian Margin station 99-6 (Epping et al., 2002), which offers one of the most complete sets of constraints for organic matter kinetics and porewater chemistry for the chemical species included in the paper. This site was chosen because (i) it lies in a coastal margin environment, (ii) porewater profiles of $O_2$, $NO_3^-$ and $NH_4^+$ are available, and (iii) the OM kinetics and flux are well constrained compared to many other coastal datasets. The other locations in this paper were either in the deep ocean, lacked porewater profiles, or combined information from multiple stations (e.g. the North Sea compilations).

We tuned only the organic matter parameters that are directly constrained by the observations: the FPOC flux and the total degradation rates (k_fast + k_slow, and made sure they match the reactivity supported from in-situ O2 microprofiles reported in Epping et al. (2002)). We did not perform any additional optimization to force the profiles to match the observed porewater data.

Physical settings such as bioturbation, irrigation and the global carbonate kinetics were kept at as universal parameters in RADIv2 and not adopted from Epping et al. (2002). We disabled dispersion as there was no permeability data available to constrain this, and dispersion led to too high $O_2$ pumping to the sediment, compared to observations.

RADIv2 reproduces the main structure of the observed $NO_2^-$, $NH_4^+$ $O_2$ profiles. We have added a new figure and a short description in the supplementary data ton this comparison and to clarify both the strengths and limitations of RADIv2 in this coastal setting.

In anticipation of the reviewer's follow-up question regarding vertical resolution, we also performed a sensitivity test in which we repeated the Iberian Margin 99-6 simulation with different uniform layer thicknesses in the upper sediment (dz = 1 mm, 2 mm, 5 mm).

The pore-water Figure and description are now added in the supplementary material (Sect. S1, Fig. S1). We refer to this in the text in **L323:** In addition, we illustrate the performance of RADIv2 for resolving depth profiles of porewater composition at a coastal margin station on the Iberian Margin (Sect. S1, Fig. S1).

And in the subsection of the Iberian Margin starting at **L362:** A dedicated comparison of simulated and observed $O_2$, $NO_3^-$ and $NH_4^+$ porewater profiles at station 99–6 is provided in the Supplement (Sect. S1, Fig. S1).

[Figure]

**General Comments**

**RC2: I agree with the first reviewer about the statistical validation of the model derived fluxes in comparison to observations. One other thing I might add as well is the lack of detail in how the five environmental drivers used in the empirical formulation of the metamodel was selected. Was there any stepwise selection screening to diagnose what**

**variables are sensitive to benthic fluxes of DIC, O2, TA? Were the selected variables based on prior literature findings (I can't find any cross-reference in the text)?**

**AC:** We added a statistical test that quantifies whether the model shows a significant mean bias relative to the observations (via a Welch two-sample t-test) and whether the simulated fluxes fall within the bulk (25–75 %) and overall (5–95 %) range of observed values. Figures 2-5 have been updated and now show a boxplot of the site-wise fluxes and overlay points for each observational site, colour-coded by dataset, using the published site means. This representation shows where the bulk of the observational fluxes lie, how different datasets contribute to the overall spread, and how the RADIv2 distribution overlaps with (or deviates from) this cloud of observations.

We agree that the rationale for selecting the five environmental drivers in the metamodel should be described more clearly. In the revised manuscript, we now state explicitly that these drivers were chosen based on previous experimental, field, and modelling work that identifies bottom-water temperature, organic-matter and carbonate supply, saturation state, and near-bottom hydrodynamics as the dominant controls on benthic DIC, O2, and TA fluxes. **L420:** These predictors were chosen based on experimental, field, and modelling studies that identify them as the dominant physical and biogeochemical controls on benthic DIC, O2, and TA fluxes. Bottom-water temperature controls molecular diffusion coefficients and the rates of redox reactions (Yuan-Hui and Gregory, 1974), the POC flux drives organic-matter degradation and calcite dissolution (Emerson and Bender, 1981), while PIC concentration and the saturation state govern carbonate dissolution and precipitation (Boudreau, 2013). The current velocity modulates SWI exchange and carbonate kinetics via its effect on diffusive boundary-layer thickness (Boudreau, 2013; Sulpis et al., 2019; Glud et al., 2007), and dispersion (McGinnis et al., 2014). Within the RADIv2 ensemble, these five drivers explain most of the variance in the simulated benthic fluxes across the six calibration sites, while keeping the metamodel compact and compatible with the variables typically available as diagnostics in GOBMs.

**RC2: Also, it will be beneficial to comment on the computational performance Radiv2 compared to version 1 as this version has exploited Julia's differential equations packages. I agree with the comment of improvement in computational efficiency of the newer version of Radi but details on how much and how well this improvement is will be useful to help guide potential users in their expectation as other competing framework are growing in this space (see SedTrace.jl – J Du, 2023).**

**AC:** We agree that comparing the computational of RADIv2 to RADIv1 is a valuable addition to the manuscript. To keep the comparison as direct as possible, we re-ran the deep-sea Southern Pacific case (input file IC_SM7) from the original RADIv1 paper with both model versions, using comparable physics (no dispersion, constant DBL, identical forcing and boundary conditions). All timings below are for a single CPU core.

For a 50-year spin-up, RADIv1 required 5 min 20 s, whereas RADIv2 (using Rosenbrock23 with a sparse KLU linear solver and abstol = $1\times10^{-6}$, reltol = $5\times10^{-3}$) completed in 40 s

For a 2000-year integration of the same set-up, RADIv1 required $\approx$ 4 h 30 min, while RADIv2 completed in $\approx$ 27 min

When we activate the new physics in RADIv2 (varying DBL and dispersion) for the same 2000-year run, the time remains of the same order ($\approx$ 28 min), showing that the added physical realism does not make the system prohibitively expensive to integrate.

The main performance gain comes from the stiff ODE integrator with adaptive time-stepping and a sparse KLU solver. RADIv1 used a forward Euler scheme with a constant time-stepping strategy, which forces the model to use this timestep, even after the solution is close to steady state. RADIv2 takes many small steps initially, but then increases the time step once the system has largely equilibrated, which is particularly beneficial for long term simulations. Moreover, using a sparse LU factorization like KLU only uses the non-zero entries of the Jacobian, which reduce memory use and computation, so stiff simulations run faster.

With reltol = $5\times10^{-3}$ and abstol = $10^{-6}$, the solver keeps the local error per timestep below about 0.5% of the current concentration for typical state-variable magnitudes.

For most Julia ODE solvers (like Rosenbrock23), the local error per timestep is kept below

allowed error=max(abstol, reltol·|y|).

This means that the solver is allowed to be wrong by at most:

either 1e-6 (a hard minimum absolute tolerance),

or 0.5% of the current value ($5\times10^{-3}$ = 0.5%), whichever is larger.

For example, at 1 mol m$^{-3}$ the allowed local error is 0.005 mol m$^{-3}$, and at $10^{-3}$ mol m$^{-3}$ it is $5\times10^{-6}$ mol m$^{-3}$. We consider this sub-percent accuracy sufficient for the equilibrium benthic fluxes, as the error will be larger in underlying assumptions than the solver. Tightening tolerances further mainly increases runtime without appreciably changing the fluxes.

We have added a new subsection "Performance of RADIv2 versus RADIv1" within the "Numerical methods and computational enhancements subection" where this comparison is described.

**Specific comments**

**RC2: L71: How does RadiV2 steady state work. Julia differentialEquations.jl can deal with steady state but depending on the dynamic itself (eg oscillating equilibrium, unstable equilibrium), it can be difficult in finding solution space that is well-posed (Steady State**

**Solvers · DifferentialEquations.jl). The paper state nothing about the numerical scheme employed in this new version of Radi that makes it different from the previous version (which I believe was a first order Euler-based scheme).**

**AC:** This is a great remark. Indeed, differentialEquations.jl can deal with steady state itself. You can change the line prob = ODEProblem(f, u0, tspan, (model_params=model_params,)) to SteadyStateProblem(f::ODEFunction, u0, p = NullParameters(); kwargs...)) in the model to make use of the build-in steady state solver. One disadvantage, as the reviewer points out, is that this can be a "black box" approach to steady state.

RADIv2 does not rely on the generic SteadyStateProblem interface of DifferentialEquations.jl and this is also not used for the runs in this paper. For the diffusive–reactive column with fixed boundary conditions considered here, we expect a unique stable steady state. Solutes equilibrate fast (years–decades), while particulate OM equilibrates much more slowly (hundreds–thousands of years) because it is buried and accumulates. So, when solid OM reaches a steady state, we assume that everything else is also at a steady state. For steady state, we require that the maximum change in the concentration of solid organic carbon (fast + slow) over the final 5 years of integration is below $10^{-3}$ mmol m$^{-3}$ across the full depth of the simulated sediment column. We now describe this explicitly in the revised manuscript in **L324:** Ensemble simulations were integrated with the stiff Rosenbrock23 solver to steady state, using adaptive time stepping and a sparse linear solver. Dissolved species equilibrate on time scales of years to decades, whereas particulate organic matter adjusts more slowly as it is buried. We therefore defined steady state by requiring that the maximum change in depth-integrated solid organic carbon (fast + slow pools) over the final 5 years of integration was less than $10^{-3}$ mmol m$^{-3}$ across the full depth of the sediment column.

We now explicitly state that both options for steady state are available in **L71:** Steady states can be obtained either by calling the steady-state solvers in the DifferentialEquations.jl framework or by time-integrating the transient system until a user-defined convergence criterion is met.

Due to the flexibility of DifferentialEquations.jl, different problem types can be paired with appropriate solvers depending on their stiffness. In the simulations presented in the paper, we found that the stiff Rosenbrock23 with adaptive time stepping and a sparse linear solver was efficient for these initial conditions. However, other different environments could perform better using different solvers. The setup used is described in more detail in our response on the numerical scheme and the RADIv1–RADIv2 performance comparison.

**RC2: L100: Is the dz constant, if so, what interval is it? If it is constant dz over the depth domain, then the resolution of dz is important if RadiV2 is employed in coastal sediment as processes operating in the upper few cm of the sediment drive the computed benthic fluxes. Thus, not properly resolving the underlying diagenetic processes at finer spatial scale will be detrimental to the predicted benthic flux across SWI.**

**AC:** In the standard configurations used in this study, we use a uniform vertical grid with a spacing of dz = $5\times10^{-3}$ m, which is a user-defined input in the setup files. We consider this dz a compromise between accuracy and computational cost for the global set of applications presented here, matching the range of fluxes in the observations.

To assess the sensitivity to vertical resolution, we performed a dedicated test for the Iberian Margin coastal station 99–6,for which well-resolved porewater profiles and inputs for to constrain RADIv2 are available. In this test we repeated the simulation with dz = 1 mm, 2 mm and 5 mm, and compared both the porewater profiles. This Figure is added to the supplementary material (Sect. S1, Fig. S1). In these tests, O2 and NO3− profiles remain relatively similar across resolutions, while coarser grids mainly affect the deeper part of the NH4+ profile.

This is now mentioned explicitly in **L324:** In all configurations used in this study, we employ a uniform vertical grid with a spacing of dz=$5\times10^{-3}$ m. A sensitivity test for different dz values for the Iberian Margin coastal is presented in the supplement material (Sect. S1, Fig. S1).

We fully agree that future applications would benefit from, ideally, a non-uniform vertical grid, particularly in strongly stratified coastal sediments or in simulations that require a deeper sediment column while keeping computational cost low. RADIv2 is a "living" model, and in the revised manuscript, we now note that future RADIv2 developments focus on supporting non-uniform grids (for example, with finer dz near the SWI and coarser spacing at depth) to further improve the resolution of near-surface processes. **L466:** Fluxes across the sediment–water interface are controlled by processes that are strongly concentrated in the upper centimetres of the sediment. The vertical grid spacing must be sufficiently fine to resolve the associated reaction and concentration gradients, which can be computationally costly. Future applications of RADIv2 could benefit from a non-uniform vertical grid, with finer spacing near the sediment–water interface and coarser spacing at depth, particularly in coastal sediments or simulations that require a deeper sediment column. RADIv2 is a living model, and ongoing developments focus on supporting non-uniform grids (for example, with finer dz near the SWI and coarser spacing at depth) to further improve the efficiency of simulating of near-surface processes.

**RC2: Table 1: My impression of RadiV2, is that it can simulate coastal early diagenetic processes. However, in these settings, the role of anaerobic pathway featuring metals (Iron in particular) coupled to the carbon cycle is strong. This will necessitate modelling secondary redox processes relating to metals (Iron and Manganese, sulfide and its minerals). In iron rich sediment, for example there can be significant routing of reduced sulfide to mineral phase (e.g pyrite) which can be either buried or undergo redox oscillation and ultimately alter alkalinity budget in marine sediment (Canfield, D. E., et al. (2005)). Is there a justification why this is omitted redox pathways are not included in the model?**

**AC:** We agree with the reviewer that secondary redox processes involving Fe, Mn and sulfide minerals can play a significant role in coastal sediment biogeochemistry.

In the present RADIv2 configuration we do include dissolved Fe(II) and Mn(II) and their oxidised solid phases (Fe(III) oxyhydroxides and Mn oxides), and we include their role as electron acceptors in organic matter degradation. However, we do not yet resolve a full sulfide–mineral network. Sulfide produced by sulfate reduction is consumed via simple pathways (oxidation and precipitation). As a consequence, RADIv2 in its current form is better suited to capturing the first-order partitioning between aerobic and anaerobic remineralisation, than to closing Fe–S–TA budgets in iron-rich and bioturbated coastal sediments.

This limitation is identified as a target for future model development and stated in the Discussion starting at **L467:** Both RADIv1 and RADIv2 are optimised for simulating carbon and oxygen cycling. In the present configuration, RADIv2 includes dissolved Fe(II) and Mn(II) and their oxidised solid phases, and represents their role as electron acceptors in organic-matter degradation. However, it does not resolve a full Fe–Mn–S mineral network. Sulfide produced by sulfate reduction is treated with simplified oxidation and precipitation terms (see Table 1), and processes such as explicit FeS/pyrite formation, burial and re-oxidation are not included. These missing pathways can be important for the alkalinity budget in iron-rich, bioturbated coastal sediments, because they control whether sulfide is buried as pyrite or repeatedly re-oxidised and recycled back to sulfate through Fe–S redox cycling (these alternative pathways have different net TA yields (Canfield et al. 2005)). Future extensions of RADIv2 should focus on implementing more comprehensive iron and manganese cycles with explicit sulfide–mineral pathways, following formulations such as (Fossing et al., 2004) and (Canfield et al. 2005), to improve applications in such settings.

**RC2: Section 2.1.2, I think the organization of the paper section can be improved. Generally, the paper can provide more information on the computational cost associated with the model improvement relative to the previous version.**

**AC:** We added a RADIv1-RADIv2 comparison in the revised manuscript, see previous response. We moved section 2.1.2 to the end of the method section (now section 2.2.3), as the tests refer to the DBL thickness and the dispersion, which were not introduced in the paper yet.

**RC2: L180 Eqn 6, what does b0 mean? No mention of the term in the text and it appears again in eqn 12 (L215). My assumption is it related to surface bioturbation rate, but I might be wrong.**

**AC:** Yes, that is correct. It is the bioturbation constant coefficient at the surface from Archer et al 2002, calculated as a function of the POC rain as: $0.0232810e-4*(100*Fpoc)^{0.85}$. This is now specified in the revised manuscript starting in **L181.**

**RC2: Same for eqn 8 (L200). The terms in the equation are not clearly defined. What does C2 mean? Concentration in the second layer (or what).**

**AC:** Thank you for noticing this is a mistake in the equation. C2 should be Cw, the concentration of the bottom water. This is corrected and both terms Cw and Cz are explained in the text in **L200.**

**RC2: L258 I think the cross-reference equation numbers are wrong. Eqn 10 is related to bioirrigation and has little to do with bottom water viscosity. *v* should be substituted in eqn 14. Am I right in this assertion? If so, please fix the numbering.**

**AC:** Yes that is correct, thank you. *v* (eq 15) should be substituted in Equation 14. Also, Equation 16 is said "to be expressed in Equation 12 as", while this should be Equation 14. Both cross-references have been corrected in the revised manuscript.

**RC2: In the practical application, is U in Eqn 14 derived from the terms (*k, v, u\**) and calculated implicitly or is it defined by user. Browsing through the code posed in zenodo, it seems U is imposed by the user or application. I suppose both options are valid.**

**AC:** The current velocity (U) is defined by the user, as it is a more easy to constrain input than the friction velocity directly. From U and T the friction velocity u* is calculated following Eq. 16, which is then used calculate the DBL thickness and the dispersion.

**RC2: Table 2: It seems that the depth considered here only validates the model for continental shelf/and adjacent coastal as well as deep Ocean. For delta and shallow coastal area, how will the coverage of the ensample of tested region perform.**

**AC:** This is a good point raised by reviewer 2 and aligns with the comment from reviewer 1. The locations listed in Table 2 were selected because they (i) provide sufficient data to constrain and validate RADIv2 and (ii) jointly span the range of conditions used to train the metamodel. At this stage, we did not include deltaic or very shallow coastal sites, because with the current limited number of calibration sites a single such location would have had strong leverage on the regression coefficients, unless it was artificially given less weight in the regression. For deltaic and very shallow coastal environments, specialised site- or region-specific metamodels may be more appropriate than a single universal formulation. However, and as result, RADIv2 has not yet been systematically tested in deltaic or very shallow coastal sites, and the metamodel likewise lacks explicit representation of these settings (as is also the case for carbonate-rich tropical shelves noted by Reviewer 1).

This is a limitation of the present work, and in the revised Discussion we now clarify environmental coverage explicitly, and we highlight training on a wider, yet physically consistent, ensemble of simulations as a key avenue for future improvement, particularly for more localized applications in deltas and shallow coastal systems in **L502:** However, the present

training and validation set primarily samples temperate shelf, margin, and deep-sea settings, and does not explicitly include environments such as deltaic systems, very shallow nearshore regions, or carbonate-rich tropical shelves. Expanding the training ensemble to a wider, yet physically consistent, range of bottom-water conditions and particulate fluxes, that can also be validated with in-situ observations, represents an important future improvement, especially for more localized applications, rather than global estimates.

**RC2: Are the values for pFast/slow as well as kinetic constant literature based? from previous modelling/experimental work done in these sediments or are they arbitrarily chosen?**

**AC:** The reactivity of the OM pools can be set manually, for example based on literature values or measurements, or it can be calculated from the empirical flux relationship used in RADIv1 (tuned for deep-sea settings). We have now explicitly stated this in the revised manuscript starting from **L109:** The rate constants for the fast and slow organic carbon pools can be specified directly by the user, for example based on literature values or in-situ measurements, or they can be diagnosed from the empirical flux relationship used in RADIv1 (tuned for deep-sea settings (Sulpis et al., 2022)).

For the values in Table 2, wherever site-specific constraints are available (e.g. the Iberian Margin runs), we base pfast/pslow and the kinetic constants on existing modelling and experimental studies for those sediments. In other cases, the parameters are fitted so that RADIv2 reproduces the observed ranges of benthic fluxes, but this tuning is performed within the envelope of degradation rate constants compiled by Arndt et al. (2013) for 2-G/3-G diagenetic models.

**Discussion**

**RC2: In the new version of Radi, it was stated that it can perform transient simulation, however in some of these regions there can be significant variability in physical (tide, resuspension/erosion) and biogeochemical forcing (changing boundary condition) that either need to impose or modelled. Did you test some configuration of this new version of the model in simulating these transient dynamics? Perhaps for on some stations. How were the results. Perhaps including some of that in the supplementary text will be useful.**

**AC:** We agree that testing RADIv2 under explicitly time-varying physical and biogeochemical forcing is important, especially in coastal settings where tides and seasonal productivity strongly modulate benthic fluxes.

For the sites listed in Table 2 we focused on steady-state simulations. This choice was driven by the primary goal of building and the steady-state metamodel. To demonstrate the transient capabilities of RADIv2 and how it responds to time-dependent forcings, we have added a dedicated transient test case in the Supplement material. In this experiment we impose both a physical and a biogeochemical forcing, with two different frequencies. (i) An M2 tidal current

signal that modulates the diffusive boundary layer thickness and dispersion, and (ii) two prescribed phytoplankton blooms that modulate the reactive organic matter supply.

In this set-up (see Figure below), hydrodynamics exerts a clear control on the $O_2$ flux, which varies by roughly 50-60% over the tidal cycle. This magnitude of change agrees with Glud et al., (2007), finding that the $O_2$ uptake in Aarhus Bay could vary by 30% or more because of changes in the DBL. The non-linearity in $O_2$ uptake arises primarily from the dispersion term, which scales non-linearly with current speed (Equation 19 the manuscript). However, his range is much smaller than the ∼25-fold variability over tidal cycles reported by McGinnis et al. (2014), (This variability is hydrodynamically driven by dispersion and the DBL effect is not treated explicitly). Their results rely on more sophisticated turbulence and bottom-drag parameterisations than those currently implemented in RADIv2. In designing RADIv2, we deliberately limited the number hydrodynamic "dials" (e.g. roughness length, TKE dissipation, eddy size as they are hard to constrain inputs).

In addition, this experiment highlights an interesting interplay between current speed and benthic fluxes in RADIv2. Higher current velocities enhance dispersion and pump oxygen into the sediment, weakening the bottom-water–porewater gradient, while at the same time thinning the DBL. In permeable, high-energy environments, however, solute exchange across the SWI is not purely molecular diffusion but spans over different regimes (e.g. Voermans et al., 2018). In RADIv2, these regimes, and the conditions under which each dominates tracer exchange, are not yet fully included. Extending the flux formulation towards such a framework would further improve SWI exchange in permeable, high-energy settings.

During the bloom periods, the $O_2$ uptake increases by about ~20%, relative to baseline conditions. The magnitude and timing of this increase depend on the reactivity of the OM pulse: highly reactive carbon produces a stronger, near-instantaneous enhancement of $O_2$ uptake, whereas more refractory carbon is retained longer in the sediments and remineralizes gradually, leading to a more sustained increase in $O_2$ uptake over longer timescales.

We now describe this configuration and its main results in the Supplementary Text and refer to it in the manuscript starting at **L325**: Although all validation experiments in this study are run to steady state, RADIv2 also supports transient simulations with time-dependent boundary conditions and forcings. This transient configuration with tidal current variability and seasonal plankton-bloom forcing is presented in the Supplement (Sect. S2, Fig. S2).

And in **L447**: A transient test case with tidal currents (Supplement material, Sect.S2, Fig.S2) illustrates how the present formulation in RADIv2 translates such hydrodynamic forcing into time-varying benthic $O_2$ fluxes.

[Figure]

**RC2: Summary of RadiV2 performance: Here, a short sentence on the computational performance of RadiV2 as compared to the previous version would be useful.**

**AC:** In the "Summary of RADIv2 performance" section we focus specifically on how RADIv2 compares to in-situ observations and to previous published model–data estimates, which are not simulated in RADIv1. We have added a separate subsection where we present an as much as possible "apples-to-apples" computational performance comparison between RADIv2 and RADIv1 for identical test cases. There we report the relative speed-up and numerical behaviour of RADIv2 with the new stiff ODE framework and solvers, see response to previous comment.

We have also renamed this section to "Summary of RADIv2 performance against in-situ observations and to previous published model–data estimates" to avoid confusion.